# Enabling robust blue circularly polarized organic afterglow through self-confining isolated chiral chromophore

Mingjian Zeng[1], Weiguang Wang[1], Shuman Zhang[1], Zhisheng Gao[1], Yingmeng Yan[1], Yitong Liu[1], Yulong Qi[1], Xin Yan[1], Wei Zhao[1], Xin Zhang[1], Ningning Guo[1], Huanhuan Li[1], Hui Li[1], Gaozhan Xie[1], Ye Tao [1,2] ✉, Runfeng Chen [1] ✉ & Wei Huang [1,3] ✉

Creating circularly polarized organic afterglow system with elevated triplet energy levels, suppressed non-radiative transitions, and effective chirality, which are three critical prerequisites for achieving blue circularly polarized afterglow, has posed a formidable challenge. Herein, a straightforward approach is unveiled to attain blue circularly polarized afterglow materials by covalently self-confining isolated chiral chromophore within polymer matrix. The formation of robust hydrogen bonds within the polymer matrix confers a distinctly isolated and stabilized molecular state of chiral chromophores, endowing a blue emission band at 414 nm, lifetime of 3.0 s, and luminescent dissymmetry factor of ~$10^{-2}$. Utilizing the synergistic afterglow and chirality energy transfer, full-color circularly polarized afterglow systems are endowed by doping colorful fluorescent molecules into designed blue polymers, empowering versatile applications. This work paves the way for the streamlined design of blue circularly polarized afterglow materials, expanding the horizons of circularly polarized afterglow materials into various domains.

Blue circularly polarized luminescence (CPL), one of the three primary colors, is of paramount importance for the application in full-color stereoscopic displays, poly-chromatic data recording, and anti-counterfeiting as well as biological imaging and optical communications[1–14]. Varied organic blue CPL materials have been prepared including fluorescence[15–19], metal-complex[20,21] and thermally activated delayed fluorescence (TADF)[22,23] materials, which have been applied in field-effect transistors[24], organic light-emitting diodes[25], and information storage[26]. Recently, circularly polarized organic afterglow (CPOA) has gained significant attention as a cutting-edge research field due to its exceptional photophysical properties for promising applications in various domains[27–33]. To design and develop CPOA materials, chiral chain engineering[34], ionic co-crystal[35], polymerization[36–38], and

multicomponent host-guest strategies[39–41] have been proposed. Although remarkable achievements in enabling CPOA materials have been witnessed, developing blue CPOA materials with exceptionally prolonged lifetimes and effective chirality still remains a formidable challenge, let alone attaining adjustable CPOA emission colors.

Three necessary prerequisites should be met to achieve CPOA emission. Firstly, the introduction of effective chirality into luminescent phosphor chromophores is essential[42]; secondly, it is imperative to enhance triplet exciton generation by expediting intersystem crossing (ISC) from singlet to triplet excited states[43]; lastly, the stabilization of the triplet excitons of phosphors through the construction of stable and stiff molecular environments for suppressing non-radiative decay pathways is of paramount need[44]. Proverbially, triplet excitons are easily exhausted

[1]State Key Laboratory of Organic Electronics and Information Displays & Institute of Advanced Materials (IAM), Nanjing University of Posts & Tele communications, Nanjing, China. [2]Songshan Lake Materials Laboratory, Dongguan, Guangdong, China. [3]Frontiers Science Center for Flexible Electronics (FSCFE), MIIT Key Laboratory of Flexible Electronics (KLoFE), Northwestern Polytechnical University, Xi'an, Shanxi, China. ✉e-mail: iamytao@njupt.edu.cn; iamrfchen@njupt.edu.cn; iamwhuang@njupt.edu.cn

through various non-radiative transition channels including triplet-triplet annihilation and luminescence quenching[45]. A possible approach to enhance CPOA emission is through chiral crystal engineering, which involves intermolecular locking and stacking to suppress non-radiative transitions of triplet excitons[46–49]. However, this tactic that leans upon ordered molecule packing may lead to the loss of triplet excitons through triplet-triplet annihilation and result in a spectrum shift towards longer wavelengths in the afterglow emission, making the development of long-lived blue CPOA challenging. Notably, compared to the aggregated state in molecular crystal[50], the CPOA emission not only can experience a blue-shift when a single molecule is appropriately confined into a polymer matrix but also can empower the stabilization of triplet excitons for an ultralong lifetime. Based on these insights, we have proposed a strategy that involves the self-confinement of isolated chiral chromophores within a rigid polymer matrix to minimize non-radiative transitions for effectively boosting blue CPOA polymers (Fig. 1a). In this work, blue CPOA polymer demonstrates ultralong lifetimes of up to 3.0 s and maximum luminescent dissymmetry factor $|g_{lum}|$ value of $1.02 \times 10^{-2}$. More excitingly, with the aid of synergistic afterglow and chirality energy transfer (SACET), full-color CPOA polymers with color tunability are achieved through doping commercially available water-soluble fluorescent materials, showing green, red, and even white CPOA emission with a lifetime of up to 2.1 s (Fig. 1b). These results not only provide a feasible way to develop blue CPOA materials but also signify the feasibility of the SACET strategy to construct full-color CPL materials.

## Results

### Material design and synthesis

As a proof of concept, a series of CPOA polymers R/S-PAMCOOCz$_X$ (Fig. 1a) were synthesized through radical copolymerization. In this design, a pair of high triplet energy level enantiomers (Supplementary Figs. 1–15), R/S−2-((2-(9H-carbazol-9-yl) propa-noyl)oxy)thyl acrylate-with (R/S-VCOOCz)[51,52], is chosen as the blue light-emitting monomer, which simultaneously exhibits good phosphorescent properties and chirality. Polyacrylamide (PAM), which has carbonyl and amino groups is chosen as a matrix because it not only can effectively promote ISC to generate the triplet excitons but also can form a strong hydrogen-bonding network to confine the blue chromophore for inhibited the non-radiative decay of triplet excitons.

### Photophysical properties of blue CPOA polymer

To ensure the chiral purity, the chiral resolution for R-VCOOCz and S-VCOOCz were performed. The enantiomeric excess values for R-VCOOCz and S-VCOOCz are calculated to be 99.9% and 98.1% (Supplementary Fig. 14). Moreover, the calculated circular dichroism (CD) spectra (Supplementary Fig. 15 and Supplementary Data 1) of R-VCOOCz and S-VCOOCz are consistent with the experimental spectra, which confirm the absolute configuration of R-VCOOCz and S-VCOOCz. Chiral polymers R/S-PAMCOOCz$_X$ (X = 1 ~ 4) were synthesized through radical binary copolymerization using the self-designed central chiral monomer of R/S-VCOOCz and acrylamide (AM) (Supplementary Fig. 1) with molar feed ratio of 1:50 (R/S-PAMCOOCz$_1$), 1:100 (R/S-PAMCOOCz$_2$), 1:200 (R/S-PAMCOOCz$_3$) and 1:400 (R/S-PAMCOOCz$_4$). The structure characterizations of the target chiral monomer and polymers were confirmed by nuclear magnetic resonance spectroscopy, powder X-ray diffraction, and gel permeation chromatography (Supplementary Figs. 1–17 and Supplementary Table 1).

R/S-PAMCOOCz$_2$ shows carbazole-dominated absorption spectra in both solution and amorphous thin film, which exhibits π-π*

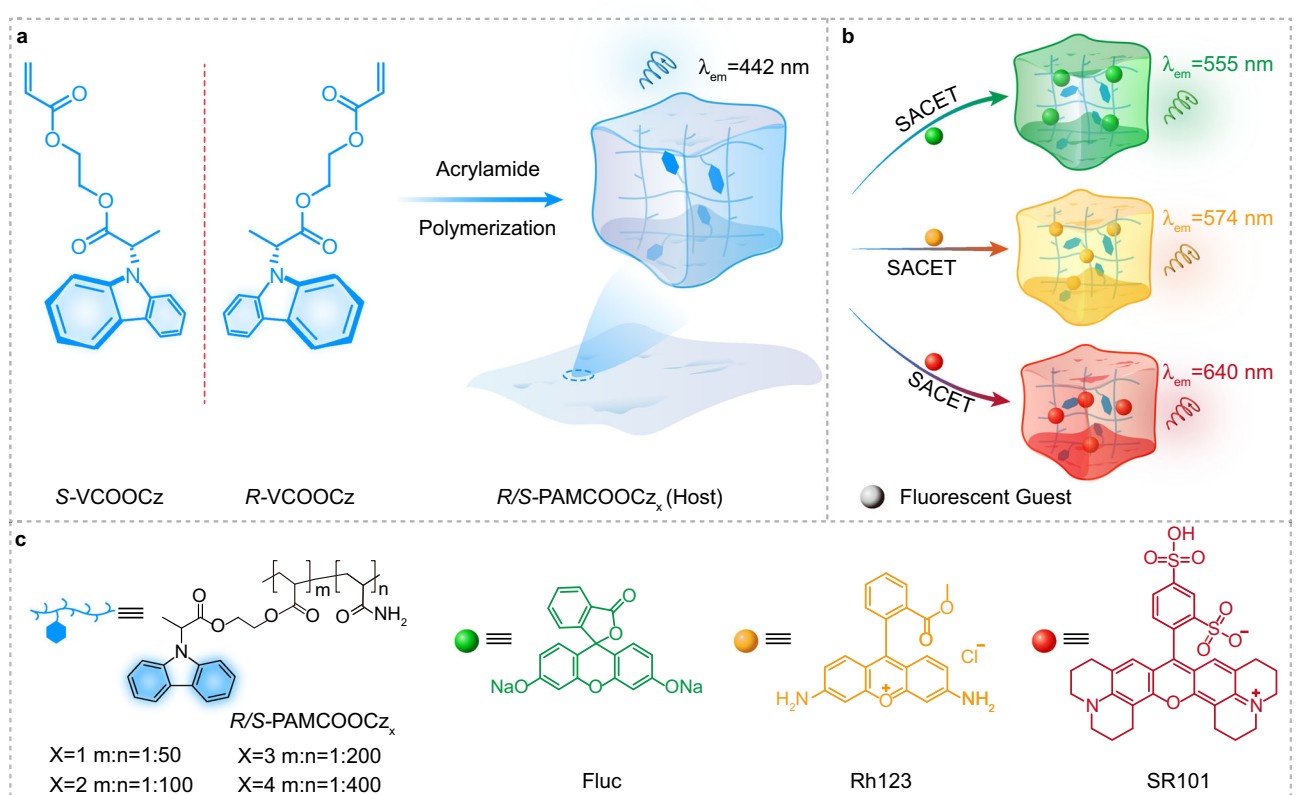

**Fig. 1 | Construction of blue CPOA polymers for enabling full-color chiral afterglow emission. a** Design of blue CPOA copolymer through covalent self-confinement of isolated chiral chromophores within a polymeric matrix. **b** Synergistic afterglow and chirality energy transfer (SACET) for achieving full-color CPOA polymers. **c** Molecular structures of PAMCOOCz$_X$ (X = 1 ~ 4), fluorescent dyes fluorescein sodium (Fluc), rhodamine 123 (Rh123), and sulforodamine (SR101).

absorption band at ~290 nm and $n$-$\pi$* absorption band near 330 nm (Supplementary Fig. 18), indicating that carbazole is the main absorption and chromophore unit[53]. Upon 254 nm UV light excitation, intense blue luminescence was recorded with emission peaks at 364 nm and lifetimes of 12.9 ns and 11.7 ns in $R$- and $S$-PAMCOOCz$_2$ films (Fig. 2a, b and Supplementary Fig. 19). Excitingly, after the cease of excitation light, the obvious ultralong-lived blue afterglow could be observed by the naked eye under ambient conditions (Supplementary Fig. 19), showcasing emission peaks at 414, 442, and 470 nm (Fig. 2a) with lifetimes of 3.0, 3.1, and 3.1 s as well as 3.0, 3.1, and 3.0 s for $R$- and $S$-PAMCOOCz$_2$ films (Fig. 2c and Supplementary Table 2), respectively. The CD spectra show mirror curves, which are consistent with the absorption spectra of $R/S$-PAMCOOCz$_2$ films, suggesting the successful introduction of chirality into the blue afterglow $R/S$-PAMCOOCz$_2$ polymers (Supplementary Fig. 20). CPL spectra reveal a strong and mirror signal (Fig. 2d, top panel) exhibiting main emission peaks at 364 nm with corresponding $g_{lum}$ values of $+6.4 \times 10^{-3}$ and $-7.8 \times 10^{-3}$, and shoulder emission peaks at 442 nm with $g_{lum}$ values of $-1.02 \times 10^{-2}$ and $+6.8 \times 10^{-3}$ for $R$- and $S$-PAMCOOCz$_2$ films (Fig. 2d, bottom panel), respectively. These results indicate the achievement of a blue CPOA polymer.

To systematically investigate the effect of molar feed ratios on the blue CPOA emission, the chiral polymers with different molar feed ratios between $R/S$-VCOOCz and AM were constructed. Considering the quite similar photophysical properties of $R$- and $S$-PAMCOOCz$_2$ films, $S$-PAMCOOCz$_X$ polymers were selected as the model polymers (Supplementary Figs. 21 and 22) to perform the investigations. The afterglow intensities and CPL signals are firstly enhanced when the molar feed ratios of $S$-VCOOCz/AM rise from 1: 50 to 1: 100; however, with further increase the feed ratio of $S$-VCOOCz/AM from 1: 100 to 1:

200 and 1: 400, the afterglow intensities and CPL signals are gradually decreased. These results suggest that the rigidity of $R/S$-PAMCOOCz$_2$ film suppresses non-radiative decay of triplet excitons, thus enabling elongated lifetime, improved afterglow intensity as well as enhanced CPL properties. It should be noted that, with further increase AM content, the hydrogen bonds in the corresponding polymeric films are largely enhanced, thus achieving identical lifetimes and non-radiaitve decay rates (Supplementary Table 3) in $S$-PAMCOOCz$_X$ (X = 1 ~ 4) films; however, the concentration of chiral chromophores $S$-VCOOCz is decreased, leading to the largely decreased afterglow intensities and CPL signals. Considering the similar DC spectra (Supplementary Fig. 22), the higher CPL intensities endow higher $g_{lum}$, thus empowering the $g_{lum}$ values in the order of $S$-PAMCOOCz$_2$ > $S$-PAMCOOCz$_3$ > $S$-PAMCOOCz$_4$ > $S$-PAMCOOCz$_1$. Therefore, $R$- and $S$-PAMCOOCz$_2$ films render the best CPOA attributes and their PLQY reached 28.6% and 24.7%, respectively (Supplementary Table 4). Notably, the copolymerization is much more effective than the physically blended polymer system of PAM and $R$-VCOOCz in endowing the CPOA emission (Supplementary Fig. 23).

In light of the excellent afterglow and CPL characters, $R/S$-PAMCOOCz$_2$ films were chosen as the model polymers to investigate the blue CPOA properties. Time-resolved emission spectra (TRES) confirm that $R/S$-PAMCOOCz$_2$ films have strong and stable afterglow luminescence (Fig. 2e and Supplementary Fig. 24). As shown in the excitation-delayed PL emission spectra (Fig. 2f and Supplementary Fig. 25), the blue CPOA could be effectively excited by UV light ranging from 210–360 nm with optimal excitation light at 299 nm. Interestingly, the excitation delayed PL spectra of emission peaks at 414, 442 and 470 nm in $R/S$-PAMCOOCz$_2$ films are quite similar, indicating that these three emission peaks (414 nm, 442 nm, and 470 nm) originate

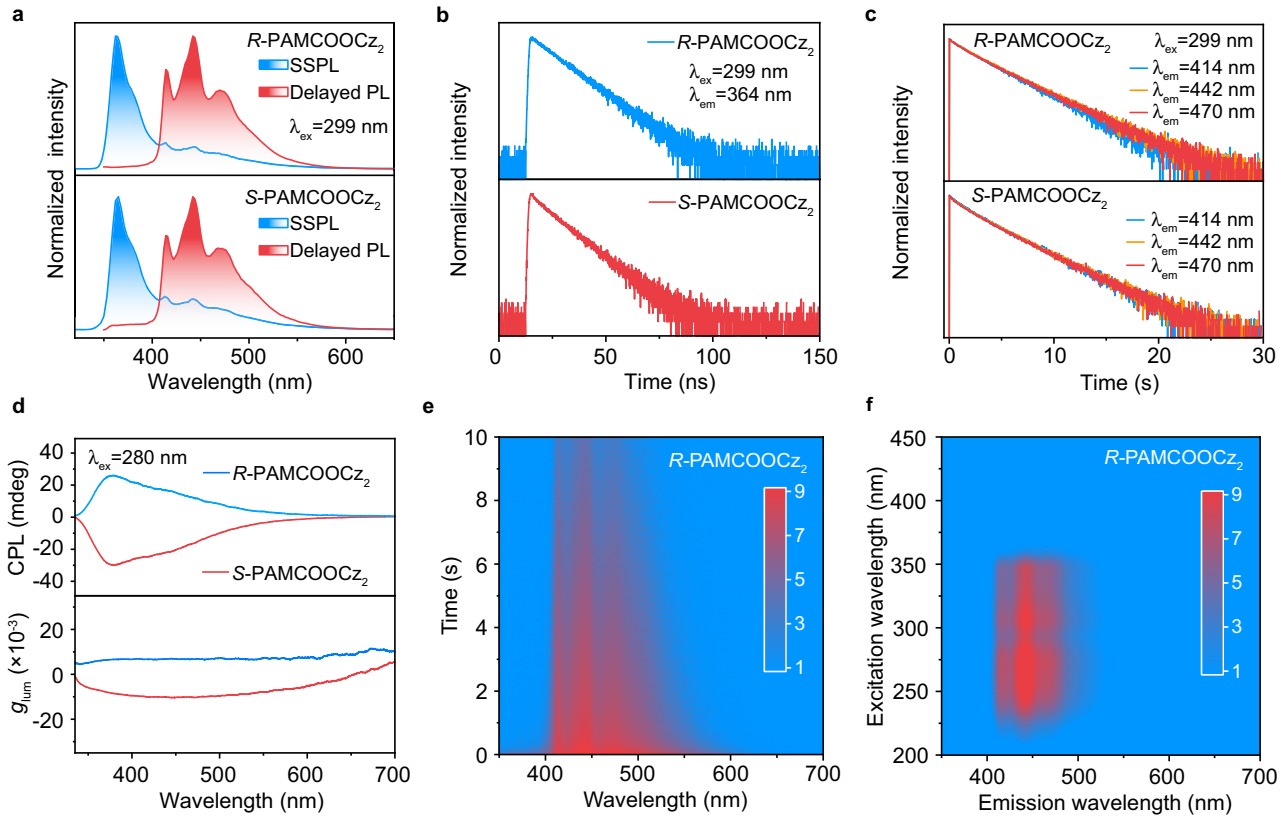

**Fig. 2 | Photophysical properties of blue CPOA polymer $R/S$-PAMCOOCz$_2$ films under ambient conditions. a** Steady-state (blue line) (SSPL) and delayed (red line) PL spectra. **b, c** Fluorescence (**b**) and afterglow (**c**) decay profiles. **d** CPL (top panel) and corresponding $g_{lum}$ curves (bottom panel). **e** Time-resolved emission spectrum. **f** Excitation-delayed PL emission spectrum.

from the same chromophore (Supplementary Fig. 26). To further investigate the luminescent source of R/S-PAMCOOCz$_2$ films, low-temperature spectra of R/S-VCOOCz monomer, R/S-PAMCOOCz$_2$, and PAM were performed (Supplementary Fig. 27). The low-temperature (77 K) delayed PL spectra of R/S-VCOOCz in dilute solution and poly-methyl methacrylate doped film are consistent with these of the R/S-PAMCOOCz$_2$ films at 77 K and room temperature, demonstrating the blue afterglow emission peaks at 414, 442, and 470 nm and high triplet energy level of 3.0 eV. Considering the combined results of the almost identical photophysical properties between R/S-PAMCOOCz$_2$ and R/S-VCOOCz and a broader emission peak at ~420 nm of PAM films, the luminescent origination of blue CPOA polymer should be the isolated chiral R/S-VCOOCz chromophore. This self-confining isolated chromophore of R/S-VCOOCz monomer is further confirmed by the wide-angle X-ray scattering measurements (Supplementary Fig. 28), showing almost identical patterns to that of PAM film. Only broader scattering peaks at 1.54 Å$^{-1}$ arising from PAM film are observed[54].

## Demonstration and investigation of SACET

Benefitting from the blue chiral afterglow emission with ultralong lifetime of 3.0 s and chirality with maximum $g_{lum}$ of $1.02 \times 10^{-2}$ as well as water-solubility, wide afterglow emission spectrum ranging from 400–550 nm and acceptable oscillator strength, R/S-PAMCOOCz$_2$ could be an ideal host platform to on-demand construct full-color CPOA polymers with robust and tunable afterglow emission[55]. Therefore, the commercialized water-soluble fluorescent dyes of fluorescein sodium (Fluc, $\lambda_{abs} = 400–480$ nm), rhodamine 123 (Rh123, $\lambda_{abs} = 450–520$ nm), and sulfo-rhodamine (SR101, $\lambda_{abs} = 520–640$ nm) were selected as guest to develop the full-color CPOA polymers due to the large spectra overlap between the afterglow spectra of energy donor R/S-PAMCOOCz$_2$ (host) and the absorption spectra of energy acceptor (fluorescent guest)[56–58], respectively. The well-overlapped spectra could maintain effective SACET from host to guest (Fig. 3a and Supplementary Fig. 29). Additionally, both the host and guest are water-soluble, leading to good compatibility for the physically blended host-guest system to shorten the distance between the donor and acceptors for further enhancing ET efficiency. Experimentally, to confirm the SACET, Fluc/R-PAMCOOCz$_2$ films with different Fluc weight concentrations were fabricated by mixing and evaporating the aqueous solution of Fluc and R-PAMCOOCz$_2$. As shown in Fig. 3b, besides the blue emission from R-PAMCOOCz$_2$, newly emerged luminescent peaks are found in both SSPL and delayed PL spectra in Fluc/R-PAMCOOCz$_2$ film. Compared to the emission peak of Fluc in aqueous solution and an inert poly (vinyl alcohol) doped film (Supplementary Fig. 30), the newly emerged luminescence peak at 555 nm in the SSPL and delayed PL spectra of the Fluc/R-PAMCOOCz$_2$ film should be from Fluc; and with rising Fluc concentrations, the intensities of emission peak at 550–650 nm are gradually increased while the emission from R-PAMCOOCz$_2$ is decreased, signifying the plausible occurrence of ET from R-PAMCOOCz$_2$ to Fluc. Compared to the SSPL emission (Supplementary Fig. 31a), the afterglow emission from R-PAMCOOCz$_2$ almost disappeared in Fluc/R-PAMCOOCz$_2$ film when the doping concentration of Fluc increased to 0.1 wt.%, demonstrating effective ET for achieving tunable afterglow emission. Eventually, the afterglow achieves a shift from blue to yellow-green (Fig. 3c).

To investigate the specific ET mechanism in depth, the lifetimes of the Fluc/R-PAMCOOCz$_2$ films were analyzed. Upon increasing Fluc concentrations from 0 to 0.1 wt.%, the lifetimes are decreased from 12.9 ns to 9.4 ns for fluorescence emission peaks at 364 nm and from 3.0 s to 2.0 s for afterglow emission peaks at 414 nm (Supplementary Fig. 31b and Fig. 3d), respectively. These results verify that the non-radiative ET process should be responsible for this ultralong afterglow emission from Fluc. The afterglow lifetimes of long-lived emission peaks at 555 nm are over 1.8 s (Fig. 3e and Supplementary Table 5). According to the measured amplitude averaged lifetime ($\tau_{amp}$) of Fluc/

R-PAMCOOCz$_2$ films at the emission peaks at 364, and 414 nm, the fluorescence and afterglow ET efficiencies are calculated to be 27.1% and 64.3%, respectively (Supplementary Table 6). Compared to the fluorescence ET efficiency, much-enhanced afterglow ET efficiency should be due to its larger spectra overlap between the afterglow emission of R-PAMCOOCz$_2$ and the absorption spectrum of Fluc. Moreover, the TRES of Fluc/R-PAMCOOCz$_2$ film shows continuous and pronounced luminescence with elongating the delayed time (Supplementary Fig. 32), suggesting the stability of the sensitized ultralong afterglow luminescence from Fluc, demonstrating that the ET strategy should be an alternative way to modulate the afterglow color[59]. With increase Fluc concentrations, the photoluminescence quantum efficiencies of the Fluc/R-PAMCOOCz$_2$ system are also increased from 23.3% to 28.1% (Supplementary Table 4). The excitation-delayed PL spectra of R-PAMCOOCz$_2$ and 0.1 wt.% Fluc/R-PAMCOOCz$_2$ (Fig. 3f) are almost identical, showing quite similar excitation-delayed PL spectra with effective excitation wavelength from 208 nm to 362 nm. In contrast, no afterglow luminescence can be observed when the excitation wavelength ranges from 400 nm to 450 nm where Fluc exhibits strong absorption abilities, indicating that the ultralong lifetime triplet excitons of Fluc/R-PAMCOOCz$_2$ films should be attributed to the afterglow ET from R-PAMCOOCz$_2$ to Fluc rather than the direct excitation of Fluc. The Fluc/S-PAMCOOC$_2$ systems endow similar spectra variations when modulated the doping concentrations of Fluc.

Since the Fluc/R-PAMCOOCz$_2$ and Fluc/S-PAMCOOCz$_2$ films empower effectual fluorescence and afterglow ET process, their chirality energy transfer properties were subsequently investigated. In comparison with the R-PAMCOOCz$_2$ film, a new peak at 555 nm is observed after excitation by a 280 nm light. With increased Fluc concentrations in Fluc/R-PAMCOOCz$_2$ film, the enhanced CPL signals originating from 555 nm are found while the CPL signals of R-PAMCOOCz$_2$ film at the regions of 330–450 nm are gradually decreased (Fig. 3g and Supplementary Fig. 33a). After the aiding of 0.1 wt.% Fluc, the CPL signal of R/S-PAMCOOCz$_2$ almost disappears, and only a strong CPL signal at 555 nm is retained with $g_{lum}$ values of $+ 3.4 \times 10^{-3}$ and $- 5.7 \times 10^{-3}$, respectively (Fig. 3h, i and Supplementary Fig. 33b). Namely, the chirality of R/S-PAMCOOCz$_2$ is successfully transferred to Fluc. These results agree well with the variation of SSPL emission of Fluc/R-PAMCOOCz$_2$ films (Supplementary Fig. 31a). Notably, the CD spectra (Supplementary Fig. 34) of Fluc doped R/S-PAMCOOCz$_2$ films are similar to the corresponding CD spectra of R/S-PAMCOOCz$_2$ films (Supplementary Fig. 20) and the use of strong absorption peak at 460 nm of Fluc as the excitation wavelength (Fig. 3a) could not trigger the CPL emission of Fluc/S-PAMCOOCz$_2$ (Supplementary Fig. 35), confirming that the chirality of Fluc doped systems originates from the chiral R/S-PAMCOOCz$_2$ hosts[60–62].

## Enabling multicolor CPOA polymers

To further demonstrate the university of the SACET to obtain wide-range color tenability[16], the achiral Rh123 and SR101 were chosen as fluorescence guests to prepare thin films since their absorption spectra endow favorable spectra overlaps with afterglow emission of R/S-PAMCOOCz$_2$ films. As anticipated, orange and red chiral afterglow emission peaked at 574 nm and 640 nm, corresponding to the luminescence peaks of Rh123 and SR101, are observed in 0.1 wt.% Rh123 and SR101 doped R-PAMCOOCz$_2$ films, respectively (Fig. 4a and Supplementary Fig. 36). Compared to pure R-PAMCOOCz$_2$ film, significantly reduced delayed PL intensities and lifetimes are found in the emission peaks at 414 nm in Rh123 and SR101 doped films (Fig. 4b), proving the occurrence of afterglow ET. This ET can be further verified by excitation-delayed PL spectra and emission mapping, which show quite similar excitation wavelength to that of the R-PAMCOOCz$_2$ host (Fig. 4e, f and Supplementary Fig. 37). Since the doping concentration of the guest is vital to enable multicolor CPOA, the SR101 doped R-PAMCOOCz$_2$ films with the weight concentrations of 0.05 wt.%, 0.1

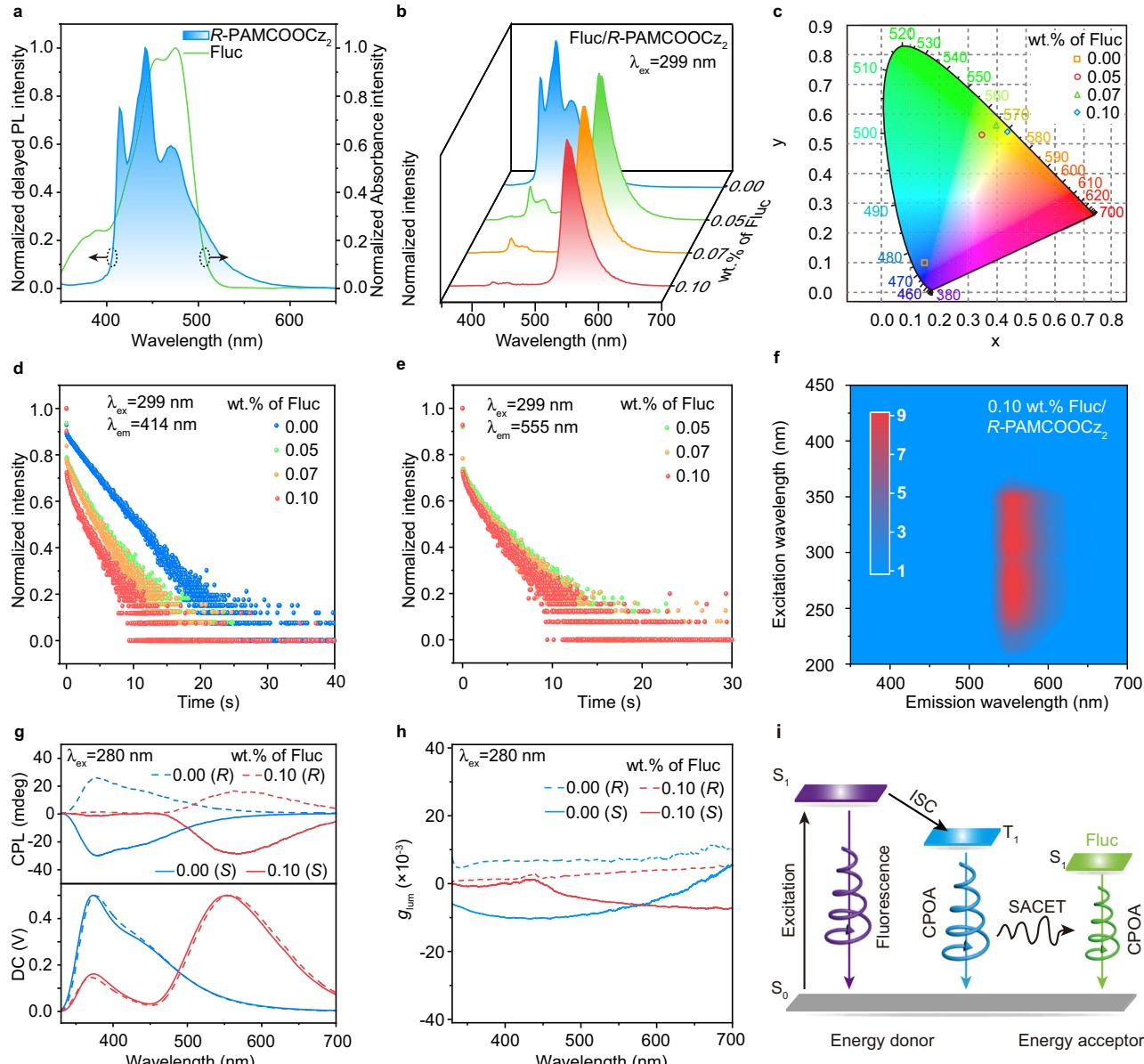

**Fig. 3 | Demonstration and investigation of SACET. a** Delayed PL spectra of *R*-PAMCOOCz$_2$ film (delayed time, 10 ms) and absorption spectra of aqueous solutions ($10^{-5}$ mol/L) of Fluc. **b–e** Delayed PL spectra (**b**), Commission International de L'Eclairage (CIE) 1931 (**c**), lifetime decay profiles at 414 nm (**d**) and 555 nm (**e**) of Fluc/*R*-PAMCOOCz$_2$ at different concen-trations. **f** Excitation-delayed PL emission spectrum of 0.1 wt.% Fluc/*R*-PAMCOOCz$_2$ film. **g, h** CPL spectra (**g**) and $g_{lum}$ values (**h**) of Fluc/*R*-PAMCOOCz$_2$ films with doping concentrations of 0.0 wt.% and 0.1 wt.%. **i** Schematic diagram of the SACET mechanism.

wt.%, and 0.2 wt.% were prepared; As shown in Supplementary Fig. 38, 0.1 wt.% SR101/*R*-PAMCOOCz$_2$ film demonstrates much enhanced SSPL and afterglow emission that originates from SR101, suggesting that the optimal doping concentration is 0.1 wt.%. Moreover, 0.1 wt.% Rh123 and SR101 doped *R*-PAMCOOCz$_2$ films obtain strong and stable long afterglow luminescence (Supplementary Fig. 32) and show ultralong lifetimes of 1.9 and 2.2 s (Fig. 4c, d and Supplementary Table 7).

Expectedly, when 0.1 wt.% Rh123 and SR101 were doped into *R/S*-PAMCOOCz$_2$ films, the CPL signal originated from *R/S*-PAMCOOCz$_2$ is largely decreased, while newly emerged emission bands derived from Rh123 (574 nm) and SR101 (640 nm) are observed (Fig. 4g, h), respectively. The maximum $g_{lum}$ of $+2.1 \times 10^{-3}$ and $-5.5 \times 10^{-3}$, $+2.5 \times 10^{-3}$, and $-3.7 \times 10^{-3}$ for 0.1 wt.% Rh123 and SR101 doped *R/S*-PAMCOOCz$_2$ films could be also recorded (Fig. 4i and Supplementary Fig. 39), figuring out the effective chirality transfer from *R/S*-PAMCOOCz$_2$ to fluorescent guests. Compared to *R/S*-PAMCOOCz$_2$ and Fluc doped *R/S*-

PAMCOOCz$_2$ films, the reduced CPL signals of 0.1 wt.% Rh123 and SR101 doped *R/S*-PAMCOOCz$_2$ films should be due to their decreased energy transfer efficiencies (Supplementary Table 6)[63]. To further verify the vital role of SACET to achieve multicolor CPOA system, the direct excitation of the achiral guest SR101 in 0.1 wt.% SR101/*R*-PAMCOOCz$_2$ films using the corresponding maximum absorption band at 550 nm as excitation light was performed. No obvious CPL signals are detected (Supplementary Fig. 40), testifying that the chiral characteristics should be derived from blue *R/S*-PAMCOOCz$_2$ polymer. More excitingly, chiral white light emission can be achieved in Rh123/*R*-PAMCOOCz$_2$ films by carefully regulating the doping concentration of Rh123 (Supplementary Fig. 41).

## Potential applications of CPOA materials
Considering CPOA materials capable of ultralong lifetime, full-color tunability, and easy water processing ability, their applications in

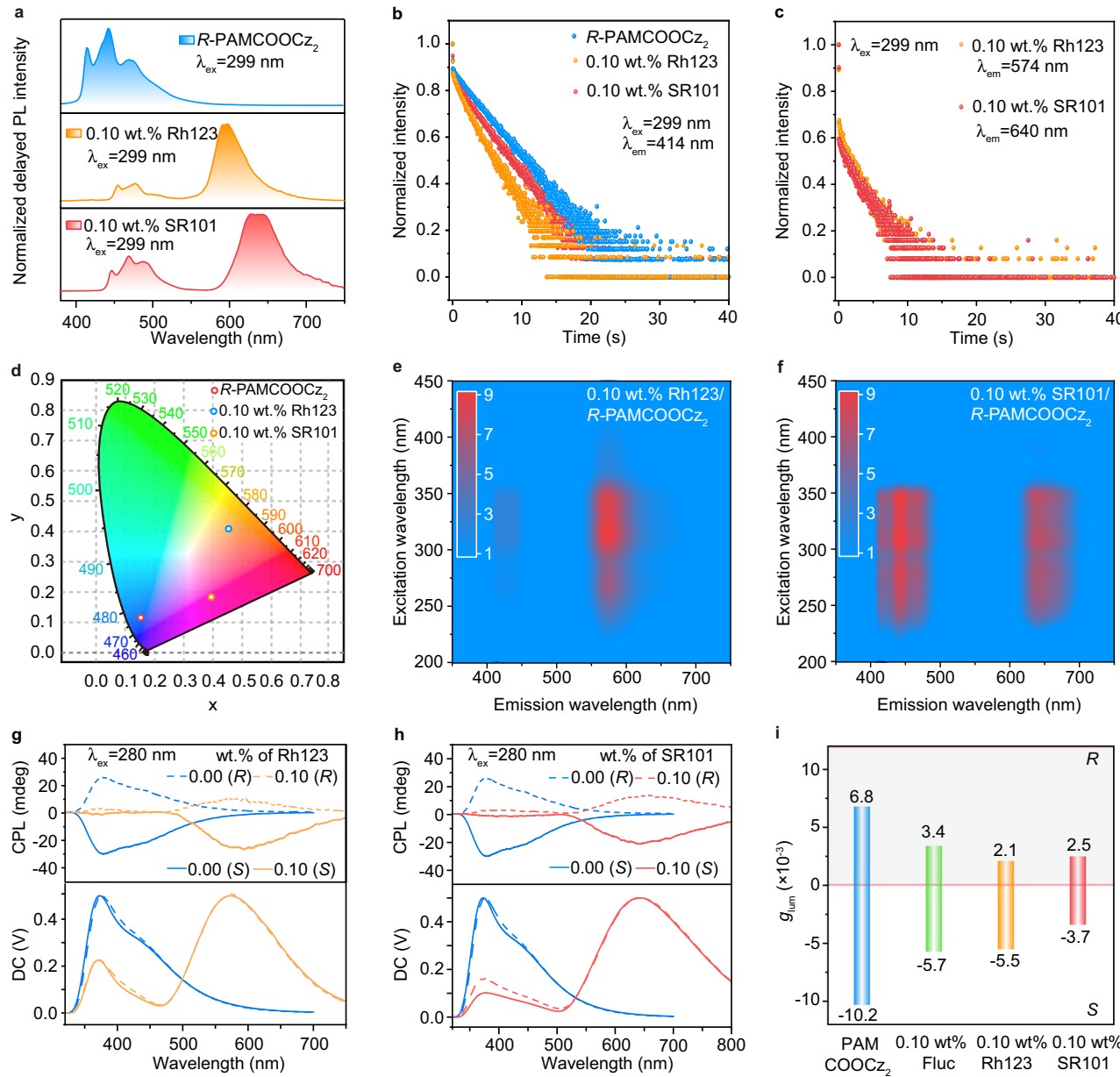

**Fig. 4 | Enabling multicolor CPOA polymers. a–d** Delayed PL spectra (**a**), lifetime decay profiles (**b, c**) and CIE 1931 (**d**) of $R$-PAMCOOCz$_2$, 0.1 wt.% Rh123/$R$-PAMCOOCz$_2$ and 0.1 wt.% SR101/$R$-PAMCOOCz$_2$. **e, f** Excitation-delayed PL emission spectrum of 0.1 wt.% Rh123/$R$-PAMCOOCz$_2$ (**e**) and 0.1 wt.% SR101/$R$-PAMCOOCz$_2$ (**f**) films. **g, h** CPL spectra of Rh123/$R$-PAMCOOCz$_2$ (**g**) and SR101/$R$-PAMCOOCz$_2$ (**h**) films with doping concentration of 0.0 wt.% and 0.1 wt.%. **i** $g_{lum}$ values of emission peaks at 440, 555, 574 and 640 nm for varied fluorescent guests doped $R/S$-PAMCOOCz$_2$ films with doping concentrations of 0.0 wt.% and 0.1 wt.% (top panel is $R$-configuration and bottom panel is $S$-configuration).

multiplex information encryption, functionalization fibers and three-dimensional objects were explored. As shown in Fig. 5a, various luminescent materials including chiral $R$-PAMCOOCz$_2$ (blue), Fluc/$R$-PAMCOOCz$_2$ (yellow-green), and SR101/$R$-PAMCOOCz$_2$ (red), and achiral PAMCz (blue), Fluc (yellow-green), Fluc/PAMCz (yellow-green), SR101 (red) and SR101/PAMCz (red) were selected as the water-soluble anticounterfeiting inks (Supplementary Fig. 42). Multiplex Morse Codes with a three-dimensional encrypted features of CPL and color, long-life had been prepared using screen-printing technology (Supplementary Fig. 43). Under daylight, due to the colorless of blue inks, the false Morse Code 1 RWHWPNK with light-yellow and pink color can be obtained. Upon irradiation by a 254 nm UV lamp, all emission colors can be observed and the false fluorescence Morse Code 2 LPHPPCQ can be recorded. After withdrawing the UV lamp, the false OA Morse

Code 3 RPSPWCM with red, green, and blue colors appeared as the Fluc and SR101 fluorescence disappeared. In contrast, with the aid of CPL analysis, the true three-dimensional encrypted CPOA Morse Code 4 showing the information of RESPECT is finally captured (Supplementary Fig. 44). Meanwhile, CPOA functionalized fibers could be easily prepared by soaking the commercial fiber into the CPOA polymer aqueous solution and then drying in an oven at 50 °C, rendering varied shapes and the tunable afterglow of blue ($R$-PAMCOOCz$_2$) and green (Fluc/$R$-PAMCOOCz$_2$) emission colors (Fig. 5b). Interestingly, colorful three-dimensional objects (Fig. 5c) emitting blue ($R$-PAMCOOCz$_2$, left), yellow-green (0.1 wt.% Fluc/$R$-PAMCOOCz$_2$, middle), and red (0.1 wt.% SR101/$R$-PAMCOOCz$_2$) CPOA emissions can be also constructed, reflecting their great potential applications in flexible wearable electronics.

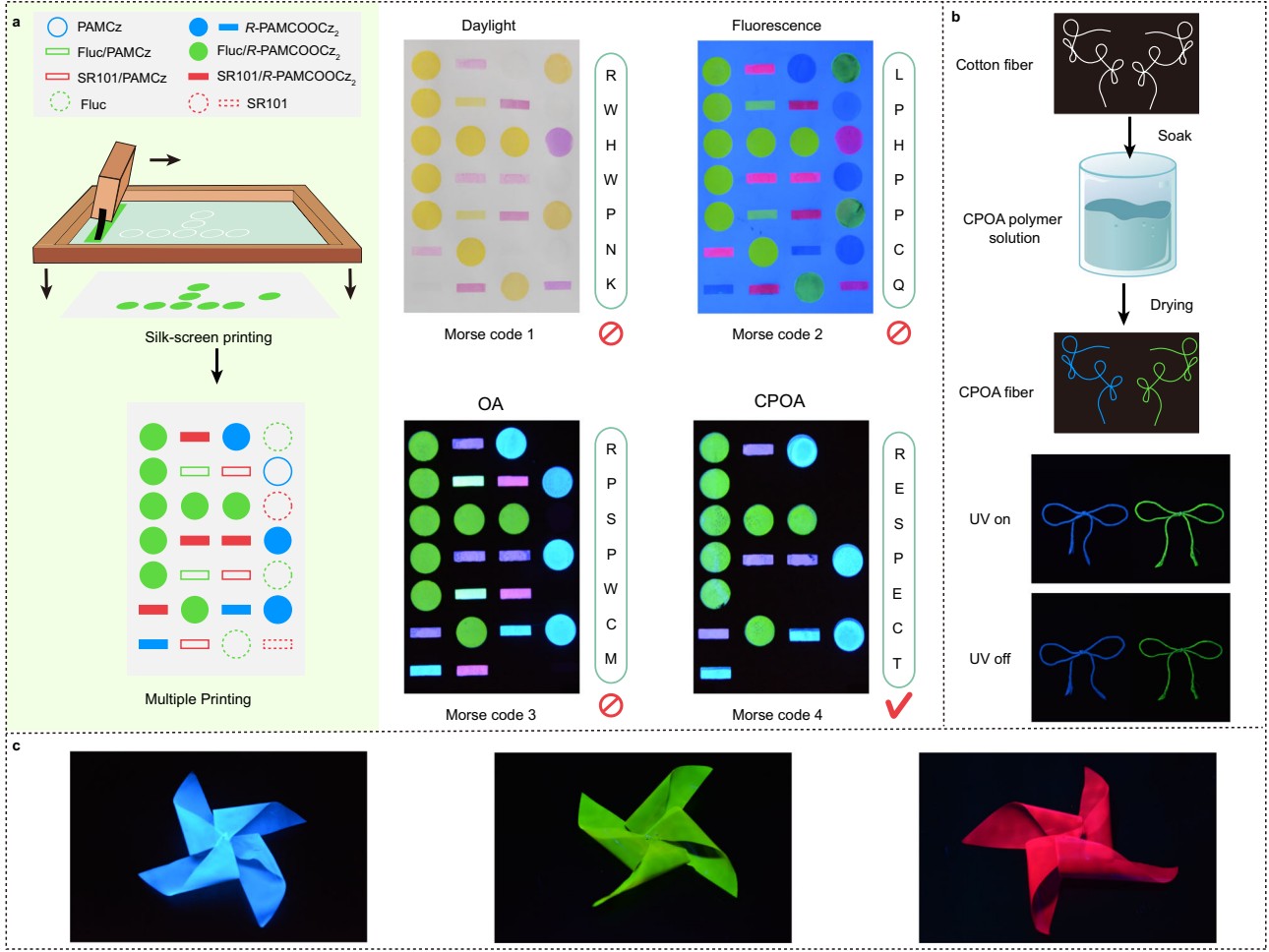

**Fig. 5 | Potential applications of CPOA materials. a** Fabrication procedure of multiplex Morse Code by screen-printing and corresponding photographs of varied Morse Code under daylight, under 254 nm UV light excitation and after turning off 254 nm UV light excitation. **b** Preparation procedures and an illustration of CPOA fibers and corresponding photographs of CPOA fibers under and after turning off 254 nm UV light excitation. **c** Blue (left), yellow-green (middle), and red (right) three-dimensional CPOA objects.

## Discussion

In summary, we have successfully proposed an efficient self-confinement method for the construction of blue CPOA polymers. This tactic depends on simultaneously confining the isolated chiral chromophores into a rigid and water-soluble polymer matrix for the stabilization of triplet excitons to enable highly efficient CPOA emission. The resultant water-soluble CPOA polymers exhibit an ultralong lifetime of up to 3.0 s, a $|g_{lum}|$ of $1.02 \times 10^{-2}$. Benefiting from afterglow and chirality properties, a series of full-color CPOA systems are prepared by physically blending water-soluble non-chiral fluorescent dyes into blue CPOA polymers through SACET. Employing the full-color tunability CPOA features, multilevel information encryption, functional fibers, and three-dimensional display objects have been fabricated. This study not only guides the design and synthesis of blue CPOA materials, but also opens a map to on-demand modulate the CPOA emission colors for varied applications.

## Methods
### Materials
All reagents and solvents were purchased from Nanjing Chemical Reagent Co. and Energy Chemical. Unless otherwise specified, these reagents and solvents were used without further purification. The purities for all purchased materials are described below: carbazole (99%), methy (R)-2-chloropropionate (98%), methy (S)-2-chloropropionate

(97%), 2-hydroxyethyl acrylate (99%), acrylamide (99.9%), fluorescein sodium (99.7%), rhodamine 123 (98%), sulfo-rhodamine (95%). And 2,2'-azobis(2-methylpropionitrile) (98%) was used after three times recrystallization.

### General procedure of radical polymerization
In an argon atmosphere, 0.01 equivalent (eq) of 2,2'-azobis(2-methylpropionitrile) (AIBN) and 1.0 eq of vinyl derivative were dissolved in 25 mL freshly distilled tetrahydrofuran (THF) under ice water. After the solid was completely dissolved, the mixture was gradually heated to 55 °C and stirred for 16 h. After the reaction, the mixture was added dropwise to 200 mL methanol to precipitate polymeric materials, then the crude product was filtered, followed by washing with petroleum ether and dichloromethane, acetone in sequence. Then the solid was dissolved in deionized water and dialyzed by a dialysis tube (molecular weight cut-off = 1000) for 72 h.

### S-PAMCOOCz₁
Following the general procedure of radical polymerization using S-VCOOCz (0.337 g, 1.0 mmol, 1.00 eq), acrylamide (AM, 3.55 g, 50.0 mmol, 50 eq), and appropriate amount of AIBN (0.0836 g, 0.51 mmol, 0.51 eq) in 25 mL freshly distilled THF to afford 3.31 g white powder polymer with a yield of 85.2%. Mn = 16817 Da; Mw = 25661 Da; PDI = 1.53.

### S-PAMCOOCz₂

Following the general procedure of radical polymerization using *S*-VCOOCz (0.169 g, 0.5 mmol, 1.00 eq), acrylamide (3.55 g, 50.0 mmol, 100 eq), and appropriate amount of AIBN (0.0828 g, 0.505 mmol, 1.01 eq) in 25 mL freshly distilled THF to afford 3.51 g white powder polymer with a yield of 94.5%. Mn = 23603 Da; Mw = 38866 Da; PDI = 1.65.

### R-PAMCOOCz₂

Following the general procedure of radical polymerization using *R*-VCOOCz (0.169 g, 0.5 mmol, 1.00 eq), acrylamide (3.55 g, 50.0 mmol, 100 eq), and appropriate amount of AIBN (0.0828 g, 0.505 mmol, 1.01 eq) in 25 mL freshly distilled THF to afford 3.56 g white powder polymer with a yield of 95.8%. Mn = 22006 Da; Mw = 38117 Da; PDI = 1.73.

### S-PAMCOOCz₃

Following the general procedure of radical polymerization using *S*-VCOOCz (0.0843 g, 0.25 mmol, 1.00 eq), acrylamide (3.55 g, 50.0 mmol, 200 eq), and appropriate amount of AIBN (0.0824 g, 0.5025 mmol, 2.01 eq) in 25 mL freshly distilled THF to afford 3.51 g white powder polymer with a yield of 96.6%. Mn = 25102 Da; Mw = 41739 Da; PDI = 1.66.

### S-PAMCOOCz₄

Following the general procedure of radical polymerization using *S*-VCOOCz (0.0421 g, 0.125 mmol, 1.00 eq), acrylamide (3.55 g, 50.0 mmol, 400 eq), and appropriate amount of AIBN (0.0822 g, 0.5013 mmol, 4.01 eq) in 25 mL freshly distilled THF to afford 3.50 g white powder polymer with a yield of 97.4%. Mn = 29095 Da; Mw = 46679 Da; PDI = 1.60.

### Preparation of full-color polymer films

0.5 g of polymer powder and a certain amount of organic fluorescent dyes were dissolved in deionized water (10 mL) followed by vigorous sonication for 10 min under ambient conditions; then the solution was poured into a flask and stirred at 60 °C for 1 h to obtain a completely transparent polymer solution; finally the mixed solution was placed in a petri dish and dried in an oven at 70 °C overnight to fabricate transparent polymer films for subsequent photophysical and morphological characterizations.

## Data availability

The data that support the plots within this paper and other findings of this study are available from the corresponding author on request. Source data are provided in this paper.

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

## Acknowledgements

This study was supported in part by the Open Research Fund of Songshan Lake Materials Laboratory (2022SLABFN16 awarded to Y.T.), the National Natural Science Foundation of China (22322106 awarded to Y.T., 62075102 awarded to H.L., 22075149 awarded to Y.T., 22105104 awarded to G.X., 22305126 awarded to H.L., and 62288102 awarded to W.H.), the Jiangsu Specially-Appointed Professor Plan (awarded to Y.T.), the China Postdoctoral Science Foundation (2023M731774 awarded to G.X.), and the Hua Li Talents Program of Nanjing University of Posts and Telecommunications (awarded to Y.T.), and Natural Science Research Start-up Foundation of Nanjing University of Posts and Telecommunications (NY222070 awarded to H.L.) We also want to thank Prof. Leyong Wang from Nanjing University for the measurements of CD spectra.

## Author contributions

M.Z., Y.T., R.C. and W.H. conceived the experiments and wrote the paper. M.Z., W.W., S.Z., Z.G. and Y.Y. were primarily responsible for the experiments. X.Z., Y.L., Y.Q., X.Y., W.Z., X.Z. and N.G. Huanhuan Li, Hui Li and G.X. measured and analyzed the photophysical properties. M.Z., W.W., S.Z. and Z.G. fabricated the applications. All authors contributed to data analyses.

## Competing interests

The authors declare no competing interests.
