## [Peer Review File · Nature Communications]

Enabling robust blue circularly polarized organic afterglow through self-confining isolated chiral chromophoreREVIEWER COMMENTS

Reviewer #1 (Remarks to the Author):

Zeng et al. presented a facile strategy to develop blue circularly polarized organic afterglow (CPOA) materials by covalently self-confining isolated chiral chromophore within a polymer matrix. The resultant polymers exhibited a long lifetime luminescence of 3.0 s, a large luminescent dissymmetry factor of ~ 10 -2, and an emission peak at ~ 414 nm under ambient conditions. Benefiting from the synergistic afterglow and chirality energy transfer, the developed polymer systems are capable of adjustable CPOA emission colors for varieties of applications. This study is well-organized and investigated, providing important insights for the development of CPOA materials with tunable emission colors. Therefore, I recommend the publication of this work in Nature Communications after minor revisions.

1. Compared to the copolymerization, can the blue CPOA emission be realized through physical mixing of PAM and R/S-VCOOCz?
2. Can the other polymers, for example, PAA, PVP, be used as the matrix to develop the blue CPOA polymers?
3. Compared to Fluc (0.1 wt.%) doped R-PAMCOOCz2 film, there is only a small spectral overlap between the absorption spectrum of SR101 and the phosphorescence spectrum of R-PAMCOOCz2, but the doping ratio of SR101 (0.1 wt.%) is the same as Fluc doped R-PAMCOOCz2 film. Therefore, I suggested the author optimize the doping ratio of SR101 to achieve much more efficient CPOA emission from SR101.
4. Is it possible to give a possible explanation for the fact that the g_{lum} values of R-configurations are all smaller than the S-configurations (Figure 4I)?
5. As described in Figure S35, chiral white light emission was achieved by doping 0.03 wt.% Rh123 into the R-PAMCOOCz2 system, where only steady-state and afterglow luminescence data were provided, and the corresponding chiral measurements are missing, please add.
6. Molecular afterglow and circularly polarized materials are hot topics in chemistry and materials. To arouse a broad interest from readership in this field, some strongly related works on recent fabrication of circularly polarized materials (Angew. Chem. Int. Ed. 2023, 62, e202302751; Angew. Chem. Int. Ed. 2017, 56, 7853) and molecular afterglow (Angew. Chem. Int. Ed. 2023, 62, e202309913; Adv. Funct. Mater. 2023, 33, 2300735) could be added as references.
7. What are the photophysical properties of the PAMCz used in Fig. 5, there is no relevant data in the text, please add.

Reviewer #2 (Remarks to the Author):

This work proposed a covalent self-constrained method to construct blue CPOA polymers by separating chiral chromophore from polymer matrix. Color fluorescence molecules are doped into blue CPOA polymer by synergistic afterglow and hand energy transfer, and the full-color CPOA system is given a

variety of uses to form a strong hydrogen bond in the polymer matrix. Furthermore, the chiral chromophore has obvious isolation and a stable molecular state. Doping color fluorescent molecules into blue CPOA polymers gives full-color CPOA systems multiple uses. This work seems very interesting. Therefore, I recommend its publication after addressing the following issues.

1. The introduction section can be more detailed. Please briefly introduce the related background work in this field.
2. In addition to the blue emission of R-P AMCOOCz2, Fluc appears a new luminescence peak in the SSPL and delayed PL spectra of the Fluc/R-P AMCOOCz2 film in Figure 3. What are the possible reasons?
3. CPL signals produced by R/SP AMCOOCz2 are significantly reduced when 0.1wt. % Rh123 and SR101 are doped into R/S-P AMCOOCz2 films in Figure 4 (G). So what is the possible reason?
4. Some recent papers related to CPL are encouraged to cite, such as *Accounts of Chemical Research* 2023, 56, 2954-2967. *Angew. Chem. Int. Ed.* 2023, 62, e202300882. *Angew. Chem. Int. Ed.* 2023, 62, e202217234. *Angew. Chem. Int. Ed.* 2022, 61, e202207028.

Reviewer #3 (Remarks to the Author):

The manuscript by Huang et al. describes a carbazole-based system modified with a polymerizable chiral chain. The resulting material shows CPL with a long emission lifetime. Interestingly, when the system is doped with fluorescein or other fluorophores, thanks to energy transfer from the carbazole triplet, again CPL with a long emission lifetime is observed. The concept behind this work is very interesting and it could merit publication in *Nature Comm* in principle. On the other hand, very important details are not discussed by the authors and potential critical points need to be clarified; therefore the work is not ready for publication. I would like to reconsider a thoroughly revised version of the manuscript.

1) The molecular materials appear to be not sufficiently characterized, especially in terms of optical purity. In the first step of the synthesis, carbazole is reacted with (enantiopure) methyl R/S-2-chloropropionate. I suppose the reaction occurs via S_N2 mechanism, that is with configuration inversion at the stereocenter, but this is not discussed. Such inversion may not be complete, therefore the optical purity of the resulting products must be checked, but this apparently has not been done by the authors. The authors are therefore invited to provide Chiral HPLC traces of the two enantioenriched compounds, as well as the racemic one, in order to determine the optical purity of their samples. By the way, how is the absolute configuration of the final material established? I guess that S-PAMCOOCz and S-VCOOCz comes from R-chloropropionate (and vice versa for R-COOCz derivatives), due to the S_N2 inversion. Please clarify. Please note that this is a key point that if not sufficiently addressed it could invalidate the rest of the work.

2) The authors show that CPL is only observed when fluorescein (and the other fluorophores) are excited

through the carbazole triplet, while it is negligible when excited directly. This is one of the 2 main point of the paper. First of all, this has been observed elsewhere (eg. in 10.1002/anie.202011745), please be sure to cite the relevant articles. Anyway, the authors call for a “chiral energy transfer”. This is not so simple, especially when ultralong times are involved, over which a coherent transfer would be difficult, see for examples the theoretical works by David Andrews. In my opinion, there are simpler mechanisms that could rationalize the data reported by the authors. Suppose that only a few fluorescein molecules interact with the carbazole, only those would be able to provide some CPL activity, being the carbazole directly linked to the chiral chain. When the system is excited through the carbazole, only those fluorescein molecules interacting directly with the carbazole (and therefore CPL-active) will be excited. On the other hand, when exciting directly the fluorescein, all the fluorescein molecules not directly interacting with the carbazole (and therefore not chirally perturbed) will be excited showing a vanishing CPL. I am not saying this is the correct explanation, I am suggesting that a “chiral energy transfer” is a very complex and controversial concept that has to be fully substantiated, otherwise other explanations have to be found.

3) “S-VCOOCz is decreased, thus leading to the largely decreased afterglow intensities and CPL signals.” I would expect weaker emitted light, and hence weaker CPL, but similar lifetimes and similar τ_{lum} factors. Please clarify.

4) Why the emission spectrum of fluorescein is red-shifted when excited at 280 nm (so through the carbazole)? I would expect that whatever the way it is excited the emission would not shift due to Kasha rule. Please explain.

5) “Notably, the CD spectra (Figure S30) of the Fluc/R/S-PAMCOOCz2 and Fluc/S-PAMCOOCz2 films are found to be almost identical to that of the corresponding R/S-PAMCOOCz2 films (Figure S18) and the use of strong absorption”. This sentence is not clear. Anyway the spectra in Fig S30 are very noisy and do not allow for any analysis. Please report better spectra if possible.

6) Please specify the excitation wavelengths in Fig 2 and in other figures as well. It would be important to show a monodimensional excitation spectrum of the fluorescein both within the PAMCOOCz2 matrix and in an inert matrix, to appreciate the ET from the carbazole. The same goes for the other fluorophores employed. If the authors excited at 280-290 nm, where fluorescein has still some residual absorption, how the direct and ET components in the static emission spectrum are distinguished?

7) A few sentences are not clear or difficult to read and need to be rephrased or clarified.

a) “...optimal excitation light at 299 nm. Interestingly, the emission peaks of 414 nm, 442 nm, and 470 nm in R/S-PAMCOOCz2 films reveal quite similar excitation spectra, experimentally indicating that these three emission peaks originate from the same chromophore”

b) “these polymers, a pair of self-designed high triplet energy level”

c) “films were fabricated through physically mixed”

d) “Moreover, this sensitized ultralong afterglow luminescence from Fluc exhibits strong and stable ultralong afterglow luminescence as revealed by TRES”

e) Please define SACET in the main text

8) In general, please avoid overemphasizing concepts: adjectives like outstanding, excellent, etc. should be kept to a minimum.

Reviewer comments:

Reviewer: #1

Comment 1: Zeng et al. presented a facile strategy to develop blue circularly polarized organic afterglow (CPOA) materials by covalently self-confining isolated chiral chromophore within a polymer matrix. The resultant polymers exhibited a long lifetime luminescence of 3.0 s, a large luminescent dissymmetry factor of $\sim 10^{-2}$, and an emission peak at ~ 414 nm under ambient conditions. Benefiting from the synergistic afterglow and chirality energy transfer, the developed polymer systems are capable of adjustable CPOA emission colors for varieties of applications. This study is well-organized and investigated, providing important insights for the development of CPOA materials with tunable emission colors. Therefore, I recommend the publication of this work in Nature Communications after minor revisions.

Author reply: Thanks for the referee's professional comments and kind recommendation of our work. We have carefully addressed the points raised by the referee.

1. Compared to the copolymerization, can the blue CPOA emission be realized through physical mixing of **PAM** and **R/S-VCOOCz**?

Author reply: We are very grateful to the reviewers for the valuable comments. According to the best ratio of copolymerization, we prepared the physically mixed films by doping **R-VCOOCz** into polymer matrix of **PAM** at the mass ratio is 1: 100. The blended **R-VCOOCz/PAM** film demonstrates a blue CPOA emission similar to that of the **R-PAMCOOCz₂** film. However, both the CPOA intensity and lifetime are significantly reduced, likely due to the poor intermixing between the water-soluble polymer matrix of **PAM** and oil-soluble **R-VCOOCz** (**Figure S23 A** and **B**). Moreover, the circularly polarized luminescent (CPL) emission of blended **R-VCOOCz/PAM** film is also decreased (**Figure S23 C**) compared to **R-PAMCOOCz₂** film. In the revised manuscript, we have involved these experimental results to prove the importance of copolymerization in achieving efficient CPOA emission. Many thanks.

Added text and figure:

Page 3 of the revised manuscript and Page S22 of the revised Supplementary Information:

Notably, the copolymerization is much more effective than the physically blended polymer

system of PAM and R-VCOCz to endow the CPOA emission (Figure S23).

Figure S23. (A) Delayed PL (10 ms delay), (B) afterglow lifetime and (C) CPL profiles of R-PAMCOOCz₂ and R-VCOCz/PAM films. Insert shows the normalized delayed PL spectra.

2. Can the other polymers, for example, PAA, PVP, be used as the matrix to develop the blue CPOA polymers?

Author reply: We thank the reviewers for the constructive suggestion. As suggested by the referee, we have performed the control experiments by copolymerization of acrylic acid (AA) (R/S-PAACOOZ) or N-vinyl-2-pyrrolidone (NVP) (R/S-PVPCOOZ) with R-VCOOZ using the molar ratio of 1:100. Both R-PAACOOZ and R-PVPCOOZ exhibit blue CPOA emission (Figure R1), but their steady-state photoluminescence (SSPL), CPOA intensities and lifetime (Figures R1 A-C) are largely decreased compared to R-PAMCOOZ₂ film. Considering the above experimental results, PAA and PVP are not good matrix to boost the CPOA emission. Thanks again.

Figure R1. (A) SSPL (top panel) and delayed PL (bottom panel, 10 ms delay), (B) normalized delayed PL (10 ms delay), (C) afterglow lifetime and (D) CPL profiles of R-PAMCOOZ₂, R-PAACOOZ and R-PVPCOOZ films.

3. Compared to **Fluc** (0.1 wt.%) doped **R-PAMCOOCz₂** film, there is only a small spectral overlap between the absorption spectrum of **SR101** and the phosphorescence spectrum of **R-PAMCOOCz₂**, but the doping ratio of **SR101** (0.1 wt.%) is the same as **Fluc** doped **R-PAMCOOCz₂** film. Therefore, I suggested the author optimize the doping ratio of **SR101** to achieve much more efficient CPOA emission from **SR101**.

Author reply: We appreciate the professional suggestion raised by the referee. In accordance with the suggestion, the **SR101** doped **R-PAMCOOCz₂** films with the weight concentrations of 0.05 wt.%, 0.1 wt.% and 0.2 wt.% were prepared. As shown in **Figure S38**, 0.1 wt.% **SR101/R-PAMCOOCz₂** film exhibits much enhanced SSPL and afterglow emission that originates from **SR101** compared to the 0.05 wt.% and 0.2 wt.% **SR101/R-PAMCOOCz₂** film. Consequently, we select the 0.1 wt.% doping concentration for the investigation in **SR101/R-PAMCOOCz₂** film. We hope our added experiments can clarify the referee's question and we have involved these discussions in the revised manuscript and Supplementary Information. Thanks.

Added text and figure:

Page 5 of the revised manuscript and Page S33 of the revised Supplementary Information:

Since the doping concentration of the guest is vital to enable multicolor CPOA, the **SR101** doped **R-PAMCOOCz₂** films with the weight concentrations of 0.05 wt.%, 0.1 wt.% and 0.2 wt.% were prepared; As shown in **Figure S38**, 0.1 wt.% **SR101/R-PAMCOOCz₂** film demonstrates much enhanced SSPL and afterglow emission that originates from **SR101**, suggesting optimal doping concentration is 0.1 wt.%.

Figure S38. (A) SSPL and (B) delayed PL (10 ms delay) spectra of **SR101/R-PAMCOOCz₂** films at different doping weight concentrations (wt.%).

4. Is it possible to give a possible explanation for the fact that the g_{lum} values of **R**-configurations are all smaller than the **S**-configurations (Figure 4)?

Author reply: We really appreciate the professional and insightful comments. Experimentally, the difference between the luminescence dissymmetry factor (g_{lum}) values of **R**-configuration (**R-PAMCOOCz₂**) and **S**-configuration (**S-PAMCOOCz₂**) is very small. The g_{lum} can be calculated by $2(I_L - I_R)/(I_L + I_R)$, where I_L and I_R refer to the intensity of left-rotated circularly polarized light (I_L) and right-rotated circularly polarized light (I_R), respectively. Therefore, a larger CPL intensity may endow higher g_{lum} . As shown in **Figure 4g**, the CPL intensity of **S**-configuration is slightly larger than that of the **R**-configuration at the peak intensities, which contributes to a larger g_{lum} for **S**-configuration. The similar phenomenon could be also found in recent publications (*Angew. Chem. Int. Ed.* **58**, 17220 (2019)). We have added the possible explanation in the revised manuscript. Many thanks.

Added text:

Page 5 of the revised manuscript:

Furthermore, the higher CPL intensities of **S-PAMCOOCz_x** than these of **R-PAMCOOCz_x** may be responsible for the larger g_{lum} values.

5. As described in Figure S35, chiral white light emission was achieved by doping 0.03 wt.% **Rh123** into the **R-PAMCOOCz₂** system, where only steady-state and afterglow luminescence data were provided, and the corresponding chiral measurements are missing, please add.

Author reply: Many thanks for the referee's professional suggestion. Following the recommendation, we have supplemented CPL spectra and g_{lum} curves for 0.03 wt.% **Rh123** doped **R/S-PAMCOOCz₂** films. Mirrored CPL signals and acceptable g_{lum} values (10^{-3}) are obtained, indicating the successful enablement of chiral white light emission. We have added the results in the revised manuscript. We extend our gratitude once again for this valuable guidance.

Updated figure:

Figure S41. (A) SSPL (top panel) and delayed PL (10 ms delay, bottom panel) spectra as well as (B) the corresponding Commission International de L'Eclairage (CIE) 1931 coordinates of 0.03 wt.% Rh123/R-PAMCOOC₂ film. (C) CPL properties of 0.03 wt.% Rh123/R-PAMCOOC₂ and Rh123/S-PAMCOOC₂ film.

6. Molecular afterglow and circularly polarized materials are hot topics in chemistry and materials. To arouse a broad interest from readership in this field, some strongly related works on recent fabrication of circularly polarized materials (*Angew. Chem. Int. Ed.* 2023, 62, e202302751; *Angew. Chem. Int. Ed.* 2017, 56, 7853) and molecular afterglow (*Angew. Chem. Int. Ed.* 2023, 62, e202309913; *Adv. Funct. Mater.* 2023, 33, 2300735) could be added as references.

Author reply: Many thanks to the referee for providing such valuable information. We have cited these articles in the revised manuscript.

Added references:

30. Nie, F. & Yan, D. P. Macroscopic assembly of chiral hydrogen-bonded metal-free supramolecular glasses for enhanced color-tunable ultralong room temperature phosphorescence. *Angew. Chem. Int. Ed.* 62, e202302751 (2023).

31. Yang, X. G., Lin, X. Q., Zhao, Y. B., Zhao Y. S. & Yan, D. P. Lanthanide metal-organic framework microrods: colored optical waveguides and chiral polarized emission. *Angew. Chem. Int. Ed.* 56, 7853 (2017).

32. Zhou, B., Qi, Z. H., Dai, M. Q., Xing, C. & Yan, D. P. Ultralow-loss optical waveguides through balancing deep-blue TADF and orange room temperature phosphorescence in hybrid antimony halide microstructures. *Angew. Chem. Int. Ed.* 62, e202309913 (2023).

33. Zhou, B. & Yan, D. P. Long Persistent Luminescence from Metal-Organic Compounds: State of the Art. *Adv. Funct. Mater.* 33, 2300735 (2023).

7. What are the photophysical properties of the **PAMCz** used in Fig. 5, there is no relevant data in the text, please add.

Author reply: We appreciate the thoughtful points raised by the referee. We have included the SSPL, delayed PL, and lifetime profiles of **PAMCz**, **Fluc/PAMCz** and **SR101/PAMCz** in the revised Supplementary Information (**Figure S43**). Notably, **PAMCz**, **Fluc/PAMCz** and **SR101/PAMCz** are blue, yellow-green, and red afterglow polymers without chirality, respectively.

Added figure:

Figure S42. (A) SSPL, (B) delayed PL (10 ms delay) and (C) afterglow lifetime decay profiles of PAMCz, Fluc/PAMCz and SR101/PAMCz films.

Reviewer #2

Comment 2: This work proposed a covalent self-constrained method to construct blue CPOA polymers by separating chiral chromophore from polymer matrix. Color fluorescence molecules are doped into blue CPOA polymer by synergistic afterglow and hand energy transfer, and the full-color CPOA system is given a variety of uses to form a strong hydrogen bond in the polymer matrix. Furthermore, the chiral chromophore has obvious isolation and a stable molecular state. Doping color fluorescent molecules into blue CPOA polymers gives full-color CPOA systems multiple uses. This work seems very interesting. Therefore, I recommend its publication after addressing the following issues.

Author reply: We are grateful for the referee's professional review and the valuable recommendations provided for our work. We have thoroughly revised the manuscript based on the referee's suggestions.

1. The introduction section can be more detailed. Please briefly introduce the related background work in this field.

Author reply: Many thanks for the thoughtful suggestions to improve the importance of our work. Various potential applications in field-effect transistors (*Device*, **1**, 100176 (2023)), organic light-emitting diodes (*Trends Chem.* **5**, 734-747 (2023)) and information storage (*Angew. Chem. Int. Ed.* **62**, e202217234 (2023)) have been added to signify the importance of chiral materials. These additions have been incorporated into the revised manuscript, and the relevant references have also been cited. Thanks again.

Added text and references:

Page 1 of the revised manuscript:

which have been applied in field-effect transistors²⁴, organic light-emitting diodes²⁵ and information storage²⁶.

Page 8 of the revised manuscript:

24. Ahn, J. et al. Chiral organic semiconducting materials for next-generation optoelectronic sensors. *Device*, **1**, 100176 (2023).

25. Wu, X. G., Yan, X. Q., Chen, Y., Zhu, W. G. & Chou, P.T. Advances in organic materials for chiral luminescence-based OLEDs. *Trends Chem.* **5**, 734-747 (2023).

26. Xu, L. et al. Visible helicity induction and memory in polyallene toward circularly polarized luminescence, helicity discrimination, and enantiomer separation. *Angew. Chem. Int. Ed.* **62**, e202217234 (2023).

2. In addition to the blue emission of **R-PAMCOOC₂**, **Fluc** appears a new luminescence peak in the SSPL and delayed PL spectra of the **Fluc/R-PAMCOOC₂** film in Figure 3. What are the possible reasons?

Author reply: We appreciate the point raised by the referee. To find the origination of the newly emerged peak, we investigated the emission of **Fluc** in aqueous solution and poly (vinyl alcohol) (PVA) doped film (**Figure S30**), which shows intense luminescent peaks at 555 nm. This result demonstrates that the newly emerged luminescence peak at 555 nm in the SSPL and delayed PL spectra of the **Fluc/R-PAMCOOC₂** film should originate from the emission of **Fluc**; and the CPOA emission should be due to the energy transfer from **R-PAMCOOC₂** to **Fluc**, which can be further verified by the enhanced luminescent intensities from **Fluc** with increase doping weight concentrations of **Fluc** in **R-PAMCOOC₂** film (**Figure 3** and **Figure S31**). We hope our explanation can clarify the referee's queries. We have added the discussion and experimental results in the revised manuscript and Supplementary Information. Many thanks.

Added text and figures:

Page 4 of the revised manuscript and Page S26 of the revised Supplementary Information:

Compared to the emission peak of **Fluc** in aqueous solution and an inert poly (vinyl alcohol) doped film (**Figure S30**), the newly emerged luminescence peak at 555 nm in the SSPL and delayed PL spectra of the **Fluc/R-PAMCOOC₂** film should be from **Fluc**.

Figure S30. SSPL and delayed PL spectra of **Fluc** in aqueous solution and doped films.

3. CPL signals produced by **R/S-PAMCOOCz₂** are significantly reduced when 0.1wt. % **Rh123** and **SR101** are doped into **R/S-PAMCOOCz₂** films in Figure 4 (G). So what is the possible reason?

Author reply: Thanks for the referee's professional and thoughtful question. According to the previous investigations (ACS Nano. **14**, 2373–2384 (2020)), the greater energy transfer efficiency leads to stronger CPL signals of the guest molecule. The g_{lum} values (**Figure 4G**) decrease in the order of **Fluc** ($g_{lum}: 5.7 \times 10^{-3}$) > **Rh123** ($g_{lum}: 5.4 \times 10^{-3}$) > **SR101** ($g_{lum}: 3.7 \times 10^{-3}$). This trend aligns with the observed energy transfer efficiency (**Fluc** > **Rh123** > **SR101**). Therefore, the slightly lower energy transfer efficiencies should be likely responsible for the reduced CPL signals in 0.1 wt. % **Rh123** and **SR101** doped **R/S-PAMCOOCz₂** films. We hope our analysis can clarify the referee's questions. We have involved these discussions in the revised manuscript. Thanks again.

Added text:

Page 5 of the revised manuscript:

Compared to **R/S-PAMCOOCz₂** and **Fluc** doped **R/S-PAMCOOCz₂** films, the reduced CPL signals of 0.1wt. % **Rh123** and **SR101** doped **R/S-PAMCOOCz₂** films should be due to their decreased energy transfer efficiencies (**Table S6**)⁶³.

4. Some recent papers related to CPL are encouraged to cite, such as Accounts of Chemical Research 2023, 56, 2954-2967. Angew. Chem. Int. Ed. 2023, 62, e202300882. Angew. Chem. Int. Ed. 2023, 62, e202217234. Angew. Chem. Int. Ed. 2022, 61, e202207028.

Author reply: We are deeply grateful for the useful and professional suggestions. We have cited these references and some recently published papers that focus on the CPL materials in the introduction parts to highlight the important role of chiral materials in various applications.

Added text and references:

Page 1 of the revised manuscript:

which have been applied in field-effect transistors²⁴, organic light-emitting diodes²⁵ and information storage²⁶.

17. Liu, N., Gao, R. T. & Wu, Z. Q. Helix-induced asymmetric self-assembly of π -conjugated block copolymers: from controlled syntheses to distinct properties. *Acc. Chem. Res.* **56**, 2954-2967 (2023).

18. Xu, X. H. et al. Precise synthesis of optically active and thermo-degradable poly(trifluoromethyl methylene) with circularly polarized luminescence. *Angew. Chem. Int. Ed.* **62**, e202300882 (2023).

19. Wang, C., Xu, L., Zhou, L., Liu, N. & Wu, Z. Q. Asymmetric living supramolecular polymerization: precise fabrication of one-handed helical supramolecular polymers. *Angew. Chem. Int. Ed.* **61**, e202207028 (2022).

26. Xu, L. et al. Visible helicity induction and memory in polyallene toward circularly polarized luminescence, helicity discrimination, and enantiomer separation. *Angew. Chem. Int. Ed.* **62**, e202217234 (2023).

Reviewer #3

Comment 3: The manuscript by Huang et al. describes a carbazole-based system modified with a polymerizable chiral chain. The resulting material shows CPL with a long emission lifetime. Interestingly, when the system is doped with fluorescein or other fluorophores, thanks to energy transfer from the carbazole triplet, again CPL with a long emission lifetime is observed. The concept behind this work is very interesting and it could merit publication in Nature Comm in principle. On the other hand, very important details are not discussed by the authors and potential critical points need to be clarified; therefore the work is not ready for publication. I would like to reconsider a thoroughly revised version of the manuscript.

Author reply: We appreciate the referee's important and constructive comments, which has been instrumental in improving the quality and impact of our work. We have carefully addressed the points raised by the referee.

1) The molecular materials appear to be not sufficiently characterized, especially in terms of optical purity. In the first step of the synthesis, carbazole is reacted with (enantiopure) methyl *R/S*-2-chloropropionate. I suppose the reaction occurs via S_N2 mechanism, that is with configuration inversion at the stereocenter, but this is not discussed. Such inversion may not be complete, therefore the optical purity of the resulting products must be checked, but this apparently has not been done by the authors. The authors are therefore invited to provide Chiral HPLC traces of the two enantioenriched compounds, as well as the racemic one, in order to determine the optical purity of their samples. By the way, how is the absolute configuration of the final material established? I guess that **S-PAMCOOCz** and **S-VCOOCz** comes from *R*-chloropionate (and vice versa for **R-COOCz** derivatives), due to the S_N2 inversion. Please clarify. Please note that this is a key point that if not sufficiently addressed it could invalidate the rest of the work.

Author reply: Many thanks for the professional and thoughtful suggestions. We agree with the referee's observation that the first step synthesis mainly occurs via S_N2 mechanism. To ensure the chiral purity, we perform the chiral resolution for **R-VCOOCz** and **S-VCOOCz** in our experiments. The optical purity of **R-VCOOCz** and **S-VCOOCz** are experimentally confirmed by the chiral HPLC measurements. As shown in **Figure S14**, the enantiomeric excess (ee) values for

R-VCOOCz and **S-VCOOCz**, used in the polymerization, are calculated to be 99.9% and 98.1%. These values indicate sufficient purity for subsequent investigations into the photophysical properties of CPOA polymers. Additionally, the calculated CD spectra (**Figure S15**) of **R-VCOOCz** and **S-VCOOCz** are consistent with the experimental spectra, which confirm the absolute configuration of **R-VCOOCz** and **S-VCOOCz** (*Curr. Org. Chem.* **14**, 1678 (2010)). We have included the corresponding discussions, chiral HPLC measurements and calculated CD spectra in the revised manuscript and Supplementary Information. We hope these additional results can clarify the referee's concerns. Thanks again.

Added text and figures:

Page 2 of the revised manuscript and S15 of the revised Supplementary Information:

To ensure the chiral purity, the chiral resolution for **R-VCOOCz** and **S-VCOOCz** were performed. The enantiomeric excess values for **R-VCOOCz** and **S-VCOOCz** are calculated to be 99.9% and 98.1% (**Figure S14**). Moreover, the calculated circular dichroism (CD) spectra (**Figure S15**) of **R-VCOOCz** and **S-VCOOCz** are consistent with the experimental spectra, which confirm the absolute configuration of **R-VCOOCz** and **S-VCOOCz**.

Figure S14. HPLC profiles of racemic **VCOOCz**, **R-VCOOCz** and **S-VCOOCz**.

Figure S15. Experimental (top panel) and calculated (bottom panel) CD spectra of *R/S*-VCOOCz.

Page S3 of the revised manuscript:

Chiral high performance liquid chromatography (HPLC) measurements were performed on Shimadzu LC-20AT using CHIRALCEL OJ-H column and methanol as mobile phase.

2) The authors show that CPL is only observed when fluorescein (and the other fluorophores) are excited through the carbazole triplet, while it is negligible when excited directly. This is one of the 2 main point of the paper. First of all, this has been observed elsewhere (eg. in 10.1002/anie.202011745), please be sure to cite the relevant articles. Anyway, the authors call for a “chiral energy transfer”. This is not so simple, especially when ultralong times are involved, over which a coherent transfer would be difficult, see for examples the theoretical works by David Andrews. In my opinion, there are simpler mechanisms that could rationalize the data reported by the authors. Suppose that only a few fluorescein molecules interact with the carbazole, only those would be able to provide some CPL activity, being the carbazole directly linked to the chiral chain. When the system is excited through the carbazole, only those fluorescein molecules interacting directly with the carbazole (and therefore CPL-active) will be excited. On the other hand, when exciting directly the fluorescein, all the fluorescein molecules not directly interacting with the carbazole (and therefore not chirally perturbed) will be excited showing a vanishing CPL. I am not saying this is the correct explanation, I am suggesting that a “chiral energy transfer” is a very complex and controversial concept that has to be fully

substantiated, otherwise other explanations have to be found.

Author reply: Many thanks to the referee for the valuable suggestions and thoughtful comments. The mentioned articles that focus on the circularly polarized FRET (*J. Chem. Phys.* **151**, 034305 (2019), *Angew. Chem. Int. Ed.* **60**, 222–227 (2021)) and the topic of chirality in fluorescence and energy transfer (*Methods Appl. Fluoresc.* **7**, 032001 (2019)) have been cited in revised manuscript. Notably, the realization of circularly polarized phosphorescence through simultaneous chirality and triplet-triplet energy transfer has been reported in recent literatures (*Angew. Chem. Int. Ed.* Doi: 10.1002/anie.202315362 (2023); *Small.* Doi: 10.1002/smll.202306969 (2023)). These results may support the synergistic afterglow and chirality energy transfer.

Regarding the possible mechanism raised by the referee, if the CPL-activity of **Fluc** molecules is generated through direct interaction with carbazole, this CPL activity should not disappear when the **Fluc** guest is directly excited due to the chiral additive polymers of **S-PAMCOOCz** for furnishing chiral environment (*ACS Cent. Sci.* **9**, 1409–1418(2023)); moreover, CD signals induced by the chiral environment should be also captured in **Fluc**. However, according to our added experimental results (**Figures S34-S35**), no detectable CPL upon direct excitation of the **Fluc** guest with the maximum absorption band at 460 nm and no obvious CD signals are recorded in **Fluc** guest. We hope our explanation can clarify the referee's concern. And, we truly understand the referee for the concern on the chiral energy transfer, which is a very complex and controversial concept that must be fully understood for figuring out the underlying mechanism in the future works. Many thanks.

Figure S34. CD spectra of 0.1 wt.% **Fluc**-doped **R/S-PAMCOOCz₂** films.

Figure S35. The CPL properties of 0.1 wt.% **Fluc/R-PAMCOOCz₂** and 0.1 wt.% **Fluc/S-PAMCOOCz₂** films excited by 280 nm and 460 nm blue light. Noted: compared to the film excited by 280 nm, no CPL property was found in 0.1 wt.% **Fluc/R-PAMCOOCz₂** and 0.1 wt.% **Fluc/S-PAMCOOCz₂** films when the **Fluc** was directly excited by 460 nm, suggesting that the SACET plays a vital role in conferring the CPL afterglow nature for the fluorescent guest doped **R/S-PAMCOOCz₂** films.

Added references:

60. Jessica, W. et al. 500-fold amplification of small molecule circularly polarised luminescence through circularly polarised FRET. *Angew. Chem. Int. Ed.* **60**, 222 – 227 (2021).
61. David, A. Chirality in fluorescence and energy transfer. *Methods Appl. Fluoresc.* **7**, 032001 (2019).
62. Kayn, F., David, B. & David, A. Influence of chirality on fluorescence and resonance energy transfer. *J. Chem. Phys.* **151**, 034305 (2019).

3) “**S-VCOOCz** is decreased, thus leading to the largely decreased afterglow intensities and CPL signals.” I would expect weaker emitted light, and hence weaker CPL, but similar lifetimes and similar g_{lum} factors. Please clarify.

Author reply: We thank the referee for the careful review and constructive questions.

Theoretically, the lifetime (τ_{int}^{OA}) can be calculated by the following equation:

$$\tau_{int}^{OA} = \frac{1}{k_r^{OA} + k_{nr}^{OA}} \quad (R1)$$

$$\Phi_{OA} \leq \Phi_{ISC} \leq 1 - \Phi_F \quad (R2)$$

$$k_r^{OA} \max = \frac{\Phi_{OA}}{\Phi_{ISC}^{\min} \times \tau_{int}^{OA}} \quad (R3)$$

$$k_r^{OA} \min = \frac{\Phi_{OA}}{\Phi_{ISC}^{\max} \times \tau_{int}^{OA}} \quad (R4)$$

$$k_{nr}^{OA} \max = \frac{1}{\tau_{int}^{OA}} - k_r^{OA} \min \quad (R5)$$

$$k_{nr}^{OA} \min = \frac{1}{\tau_{int}^{OA}} - k_r^{OA} \max \quad (R6)$$

where k_r^{OA} and k_{nr}^{OA} represent the radiative and non-radiative afterglow rate constant, respectively; thus, the lifetime is inverse dependence on the k_r^{OA} and k_{nr}^{OA} . According to equations (R2-R6), the k_r^{OA} and k_{nr}^{OA} can be achieved. As indicated in **Table S3**, the $k_{nr}^{OA} \max$ is much larger than the $k_r^{OA} \min$, which determinate the lifetime of afterglow materials. On the basis of the wide-angle X-ray scattering measurements (**Figure S28F**), only broader scattering peaks at 1.54 \AA^{-1} arising from **PAM** are observed in **S-PAMCOOCz_x (X=1~4)** polymer films with varied **S-VCOOCz** concentrations. This suggests that strong hydrogen intermolecular interactions provided by **PAM** are almost the same, which can effectively inhibit the non-radiative transition, thus leading to **almost identical lifetimes** in **S-PAMCOOCz_x (X=1~4)** polymer films. **Our experimental result is in line with the referee's expectations.**

Experimentally, for the measurement of CPL using JASCO CPL-300, the DC and AC can be calculated by the following equations (*Adv. Mater.* **35**, 2302279 (2023)),

$$DC = \frac{I_L + I_R}{2} \quad (R7)$$

$$AC = \frac{I_L - I_R}{2} \quad (R8)$$

Thus, the g_{lum} can be calculated by the following equation:

$$g_{lum} = 2 \frac{AC}{DC} = 2 \frac{I_L - I_R}{I_L + I_R} \quad (R9)$$

where I_L and I_R refer to the intensity of left-rotated circularly polarized light (I_L) and right-rotated circularly polarized light (I_R), respectively. Considering the similar DC spectra in **Figure S22** (It should be noted that when we measure the CPL spectra, the DC value is suggested to keep at the 0.5 V through tuning the HT Voltage), the high CPL intensities should endow high g_{lum} . The measured CPL signals of polymers are in the order of **S-PAMCOOCz₂ > S-PAMCOOCz₃ > S-PAMCOOCz₄ > S-PAMCOOCz₁**, thus enabling the g_{lum} in the order of **S-PAMCOOCz₂ > S-PAMCOOCz₃ > S-PAMCOOCz₄ > S-PAMCOOCz₁**, which is consistent with our experimental

results. We hope our explanations can clarify the referee's concern. We have added these discussions in the revised manuscripts. Thanks again.

Added text and table:

Page 3 of the revised manuscript and S20 of the revised Supplementary Information:

It should be noted that, with further increase AM content, the hydrogen bonds in the corresponding polymeric films are largely enhanced, thus achieving identical lifetimes and non-radiative decay rates (Table S3) in S-PAMCOOC_x (X=1~4) films; however, the concentration of chiral chromophores S-VCOCz is decreased, leading to the largely decreased afterglow intensities and CPL signals. Considering the similar DC spectra (Figure S22), the higher CPL intensities endow higher g_{lum} , thus empowering the g_{lum} values in the order of S-PAMCOOC₂ > S-PAMCOOC₃ > S-PAMCOOC₄ > S-PAMCOOC₁.

Table S3. Photophysical properties of S-PAMCOOC_x (X=1~4) films excited by 299 nm UV light.

Parameters	S-PAMCOOC ₁	S-PAMCOOC ₂	S-PAMCOOC ₃	S-PAMCOOC ₄
Φ_{SSPL} (%)	22.08	24.70	26.04	26.43
Φ_F (%)	8.41	9.00	11.72	12.42
Φ_{OA} (%)	13.67	15.70	14.32	14.01
Φ_{ISC}^{min} (%)	13.67	15.70	14.32	14.01
Φ_{ISC}^{max} (%)	91.59	91.00	88.28	87.58
τ_{int}^{OA} (s)	2.94	3.12	3.06	3.05
$k_r^{OA_{min}}$ (s ⁻¹)	0.05	0.06	0.05	0.05
$k_r^{OA_{max}}$ (s ⁻¹)	0.34	0.32	0.33	0.33
$k_{nr}^{OA_{min}}$ (s ⁻¹)	0.00	0.00	0.00	0.00
$k_{nr}^{OA_{max}}$ (s ⁻¹)	0.29	0.27	0.27	0.28

4) Why the emission spectrum of fluorescein is red-shifted when excited at 280 nm (so through the carbazole)? I would expect that whatever the way it is excited the emission would not shift due to Kasha rule. Please explain.

Author reply: We extend our sincere thanks to the referee for the professional questions. To double check this result, we performed the measurements once more. As shown in **Figure S35**, the similar emission spectra are found in **0.1 wt.% Fluc/R-PAMCOOC₂** film when excited by 280 nm UV light and 460 nm blue light. We are genuinely grateful to the referee for highlighting this oversight, which has significantly contributed to improving the quality of our manuscript.

Updated Figure:

Page S32 of the revised Supplementary Information:

Figure S35. The CPL properties of 0.1 wt.% **Fluc/R-PAMCOOC₂** and 0.1 wt.% **Fluc/S-PAMCOOC₂** films excited by 280 nm and 460 nm blue light. Noted: compared to the film excited by 280 nm, no CPL property was found in 0.1 wt.% **Fluc/R-PAMCOOC₂** and 0.1 wt.% **Fluc/S-PAMCOOC₂** films when the **Fluc** was directly excited by 460 nm, suggesting that the SACET plays a vital role in conferring the CPL afterglow nature for the fluorescent guest doped **R/S-PAMCOOC₂** films.

5) “Notably, the CD spectra (Figure S30) of the **Fluc/R/S-PAMCOOC₂** and **Fluc/S-PAMCOOC₂** films are found to be almost identical to that of the corresponding **R/S-PAMCOOC₂** films (Figure S18) and the use of strong absorption”. This sentence is not clear. Anyway the spectra in Fig S30 are very noisy and do not allow for any analysis. Please report better spectra if possible.

Author reply: We appreciate the points raised by the referee. To make this sentence more intuitive for the readers, we rewrite the sentence as follows: “Notably, the CD spectra (**Figure**

S34) of Fluc doped R/S-PAMCOOCz₂ films are similar to the corresponding CD spectra of R/S-PAMCOOCz₂ films (Figure S20) and the use of strong absorption.” And we have retested the CD spectra of Fluc/R-PAMCOOCz₂ and Fluc/S-PAMCOOCz₂ films, resulting in significantly enhanced and mirrored CD signals. The updated figures had been added in the revised Supplementary Information. Many thanks.

Updated text and Figures:

Page 5 of the revised manuscript and S19, 31 of the revised Supplementary Information:

Notably, the CD spectra (Figure S34) of Fluc doped R/S-PAMCOOCz₂ films are similar to the corresponding CD spectra of R/S-PAMCOOCz₂ films (Figure S20) and the use of strong absorption.

Figure S20. CD spectra of R/S-PAMCOOCz₂ films.

Figure S34. CD spectra of 0.1 wt.% Fluc-doped R/S-PAMCOOCz₂ films.

6) Please specify the excitation wavelengths in Fig 2 and in other figures as well. It would be important to show a monodimensional excitation spectrum of the fluorescein both within the PAMCOOCz₂ matrix and in an inert matrix, to appreciate the ET from the carbazole. The same goes for the other fluorophores employed. If the authors excited at 280-290 nm, where fluorescein has still some residual absorption, how the direct and ET components in the static emission spectrum are distinguished?

Author reply: We thank the referee for the kind guidance in treating figures and nice suggestions. The specification excitation wavelength been added in the updated figures in our revised manuscript and Supplementary Information.

Furthermore, according to the suggestions, the one-dimensional excitation-delayed PL emission spectra of fluorescence guests in PAMCOOCz₂ matrix and in an inert poly (vinyl alcohol) (PVA) matrix were performed by monitoring the corresponding emission bands. In the measurements, PVA is selected as a water-soluble inert matrix, which has been widely used to develop the afterglow materials (*Nat. Commun.* **12**, 2297 (2021)). Notably, no detectable afterglow emissions can be found in fluorescent guests doped PVA films due to the absence of energy transfer from the inert PVA matrix to fluorescent guests; moreover, no obvious excitation-delayed PL spectra are achieved in fluorescent guests doped PVA films (**Figure R2**). Interestingly, almost identical excitation-delayed PL spectra are observed in fluorescent guests

doped **PAMCOOC₂** films, showing excitation wavelength ranging from 220 to 400 nm; these excitation-delayed PL spectra of fluorescent guests doped **PAMCOOC₂** films are consistent with the excitation wavelength of **R-PAMCOOC₂**; these results demonstrate that the afterglow emission in fluorescent guests doped **PAMCOOC₂** films should originate from the energy transfer from **PAMCOOC₂** to fluorescent guests, which verify that the direct excitation of fluorescence guests can not boost the CPOA emission, implying that the contribution of CPOA emission through direct excitation is negligible. We hope our explanations can clarify the referee's questions. Thanks.

Figure R2. Excitation-delayed PL spectra of **R-PAMCOOC₂** film as well as varied fluorescent guests doped **R-PAMCOOC₂** and **PVA** films.

Added text and figures:

Page 5 of the revised manuscript and S33 of the revised Supplementary Information:

This ET can be further verified by excitation-delayed PL spectra and emission mapping, which show quite similar excitation wavelength to that of the **R-PAMCOOC₂** host (Figure 4E, 4F and Figure S37).

Figure S37. Excitation-delayed PL spectra of **R-PAMCOOCz₂** film as well as varied fluorescent guests doped **R-PAMCOOCz₂** films.

- 7) A few sentences are not clear or difficult to read and need to be rephrased or clarified.
- a) “... optimal excitation light at 299 nm. Interestingly, the emission peaks of 414 nm, 442 nm, and 470 nm in **R/S-PAMCOOCz₂** films reveal quite similar excitation spectra, experimentally indicating that these three emission peaks originate from the same chromophore”
- b) “these polymers, a pair of self-designed high triplet energy level”
- c) “films were fabricated through physically mixed”
- d) “Moreover, this sensitized ultralong afterglow luminescence from Fluc exhibits strong and stable ultralong afterglow luminescence as revealed by TRES”
- e) Please define SACET in the main text

Author reply: We thank the referee’s careful review and useful suggestions. As suggested by the referee, we have rewritten these sentences and carefully checked the whole article to make the description more intuitive for the readers. Thanks once again for the constructive comments, which have significantly contributed to the enhancement of our manuscript's

quality.

Updated text:

Page 3 of the revised manuscript:

a) Interestingly, the excitation delayed PL spectra of emission peaks at 414 nm, 442 nm, and 470 nm in *R/S*-PAMCOOC₂ films are quite similar, indicating that these three emission peaks (414 nm, 442 nm, and 470 nm) originate from the same chromophore.

Page 1 of the revised manuscript:

b) In this design, a pair of high triplet energy level enantiomers (Supplementary Information, **Figures S1-S15**), *R/S*-2-((2-(9H-carbazol-9-yl) propa-nyl)oxy)thyl acrylate with (*R/S*-VCOOC₂), is chosen as the blue light-emitting monomer, which simultaneously exhibits good phosphorescent properties and chirality.

Page 4 of the revised manuscript:

c) Experimentally, to confirm the SACET, **Fluc/R-PAMCOOC₂** films with different **Fluc** weight concentrations were fabricated through mixing and evaporating the aqueous solution of **Fluc** and **R-PAMCOOC₂**.

Page 5 of the revised manuscript:

d) Moreover, the TRES of **Fluc/R-PAMCOOC₂** film shows continuous and pronounced luminescence with elongating the delayed time (**Figure S32**), suggesting the stability of the sensitized ultralong afterglow luminescence from **Fluc**.

Page 1 of the revised manuscript:

e) SACET, defined as synergistic afterglow and chirality energy transfer, is introduced and explained when the term first appears in the manuscript.

8) In general, please avoid overemphasizing concepts: adjectives like outstanding, excellent, etc. should be kept to a minimum.

Author reply: Thank the referee for the valuable suggestions. We have made the changes in the corresponding part to avoid overemphasizing concepts in the revised manuscripts.

A list of changes made:

- 1) HPLC profile of racemic **VCOOCz**, **R/S-VCOOCz** were measured and added in **Figure S14**.
- 2) Experimental and calculated CD spectra of **R/S-VCOOCz** were measured and added in **Figure S15**.
- 3) Photophysical properties of **S-PAMCOOCz_x** (**X=1-4**) films excited by 299 nm UV light were measured in **Table S3**.
- 4) Delayed PL, afterglow lifetime and circularly polarized luminescent (CPL) profiles of **R-VCOOCz/PAM** films were measured in **Figure S23**.
- 5) SSPL and delayed PL spectra of **Fluc** in aqueous solution and doped films were measured in Supplementary **Figure S30**.
- 6) CD spectra of **R/S-PAMCOOCz₂** and 0.1 wt.% **Fluc**-doped **R/S-PAMCOOCz₂** films were updated in **Figures S20** and **S34**.
- 7) The CPL properties of 0.1 wt.% **Fluc/R-PAMCOOCz₂** and 0.1 wt.% **Fluc/S-PAMCOOCz₂** films excited by 280 nm and 460 nm blue light were updated in **Figure S35**.
- 8) Excitation-delayed PL spectra of **R-PAMCOOCz₂** film as well as varied fluorescent guests doped **R-PAMCOOCz₂** films were measured and added in **Figure S37**.
- 9) SSPL and delayed PL spectra of **SR101/R-PAMCOOCz₂** films at different doping weight concentrations were measured and added in **Figure S38**.
- 10) CPL properties of 0.03 wt.% **Rh123/R-PAMCOOCz₂** and **Rh123/S-PAMCOOCz₂** film were measured and added in **Figure S41**.
- 11) SSPL, delayed PL and lifetime decay profiles of **PAMCz**, **Fluc/PAMCz** and **SR101/PAMCz** films were measured and added in **Figure S42**.
- 12) The reference and figure number has been updated.
- 13) Some grammar mistakes and unclear descriptions have been also revised.

All those updates have been highlighted in the revised manuscript and revised Supplementary Information.

REVIEWERS' COMMENTS

Reviewer #1 (Remarks to the Author):

In my view, the authors have answered all the questions and revised related points, and thus this revised work can be published as it is.

Reviewer #2 (Remarks to the Author):

The authors have revised the manuscript according to the comments of the reviewers and addressed all the concerns. Thus, the revised manuscript can be accepted for publication in Nature Communications.

Reviewer #3 (Remarks to the Author):

The authors have satisfactorily addressed my main concerns. In particular, they have shown conclusive evidence of the optical purity of their materials. I would therefore recommend publication in Nature Communications, conditional to the revisions below:

- 1) "Good chirality", this is not a well defined concept. An optical property stemming from a chiral system can be good, but not chirality itself which is a binary geometrical concept (a system is either chiral or not chiral). Please rephrase.
- 2) "Furthermore, the higher CPL intensities of S-PAMCOOCzx than these of R-PAMCOOCzx may be responsible for the larger g_{lum} values". This sentence is very misleading. The two enantiomers must have exactly the same but opposite g -value. Some minor differences can derive from unavoidable experimental error or statistical fluctuations. In the authors' case, the differences are within an acceptable limit. So, please delete this sentence.
- 3) Check the newly introduced references. They appear with first names spelled out and abbreviates surnames.

Reviewer comments:

Reviewer: #1

Comment 1: In my view, the authors have answered all the questions and revised related points, and thus this revised work can be published as it is.

Author reply: We are very grateful for the recommendation!

Reviewer #2

Comment 2: The authors have revised the manuscript according to the comments of the reviewers and addressed all the concerns. Thus, the revised manuscript can be accepted for publication in Nature Communications.

Author reply: We appreciate the reviewer's recommendation of our work!

Reviewer #3

Comment 3: The authors have satisfactorily addressed my main concerns. In particular, they have shown conclusive evidence of the optical purity of their materials. I would therefore recommend publication in Nature Communications, conditional to the revisions below:

Author reply: We appreciate the reviewer's acceptance and recommendation of our work!

1) "Good chirality", this is not a well defined concept. An optical property stemming from a chiral system can be good, but not chirality itself which is a binary geometrical concept (a system is either chiral or not chiral). Please rephrase.

Author reply: Thank the referee for the valuable comments that help us to further understand the chirality. Good chirality has been changed to effective chirality in the revised manuscripts. Many thanks.

2) "Furthermore, the higher CPL intensities of S-PAMCOOCzx than these of R-PAMCOOCzx may be responsible for the larger g values". This sentence is very misleading. The two enantiomers must have exactly the same but opposite g -value. Some minor differences can derive from unavoidable experimental error or statistical fluctuations. In the authors' case, the differences are within an acceptable limit. So, please

delete this sentence.

Author reply: We thank the referee's careful review and useful suggestions. We have deleted this sentence in revised manuscripts.

3) Check the newly introduced references. They appear with first names spelled out and abbreviates surnames.

Author reply: We are very sorry for this mistake. We have carefully checked the reference and revised the mistakes. Thanks once again for the referee.

Page 10 of the revised manuscript:

60. Wade, J. et al. 500-fold amplification of small molecule circularly polarised luminescence through circularly polarised FRET. *Angew. Chem. Int. Ed.* **60**, 222 – 227 (2021).

61. Andrews, D. L. Chirality in fluorescence and energy transfer. *Methods Appl. Fluoresc.* **7**, 032001 (2019).

62. Forbes, K. A., Bradshaw, D. S. & Andrews, D. L. Influence of chirality on fluorescence and resonance energy transfer. *J. Chem. Phys.* **151**, 034305 (2019).

All those updates have been highlighted in the revised manuscript.